Substantially positive contributions of new particle formation to Cloud Condensation
Nuclei under low supersaturation in China based on numerical model improvements
Chupeng Zhang[1#], Shangfei Hai[2,12#], Yang Gao[1*], Yuhang Wang[3*], Shaoqing Zhang[4,2],
Lifang Sheng[2], Bin Zhao[5], Shuxiao Wang[5], Jingkun Jiang[5], Xin Huang[6], Shen Xiaojing[7],
Sun Junying[7], Aura Lupascu[8,9], Manish Shrivastava[10], Jerome D. Fast[10], Wenxuan
Cheng[1], Xiuwen Guo[1], Ming Chu[1], Nan Ma[11], Juan Hong[11], Qiaoqiao Wang[10],
Xiaohong Yao[1] and Huiwang Gao[1]
[1]Frontiers Science Center for Deep Ocean Multispheres and Earth System, and Key Laboratory of
Marine Environmental Science and Ecology, Ministry of Education, Ocean University of China,
and Laoshan Laboratory, Qingdao, 266100, China
[2]College of Oceanic and Atmospheric Sciences, Ocean University of China, Qingdao, 266100,
China
[3]School of Earth and Atmospheric Sciences, Georgia Institute of Technology, Atlanta, GA, 30332,
USA
[4]Frontiers Science Center for Deep Ocean Multispheres and Earth System, and Key Laboratory of
Physical Oceanography, Ocean University of China, and Laoshan Laboratory, Qingdao, 266100,
China
[5]State Key Joint Laboratory of Environment Simulation and Pollution Control, School of
Environment, Tsinghua University, Beijing, 100084 China,and State Environmental Protection
Key Laboratory of Sources and Control of Air Pollution Complex, Beijing 100084, China
[6]School of Atmospheric Sciences, Nanjing University, Nanjing, 210023, China
[7]State Key Laboratory of Severe Weather & Key Laboratory of Atmospheric Chemistry of CMA,
Chinese Academy of Meteorological Sciences, Beijing, 100081, China
[8]Institute for Advanced Sustainability Studies, Potsdam D-14467, Germany
[9]ECMWF, Bonn, 53111, Germany
[10]Atmospheric Sciences and Global Change Division, Pacific Northwest National Laboratory,
Richland, WA, 99354, USA
[11]Institute for Environmental and Climate Research, Jinan University, Guangzhou, 510000, China
[12]CMA Earth System Modeling and Prediction Center, China Meteorological Administration,
Beijing 100081, China

[#]Authors contributed equally to this study.

[*]To whom correspondence to: yanggao@ouc.edu.cn, yuhang.wang@eas.gatech.edu

**Abstract**

New particle formation (NPF) and subsequent particle growth are important sources of condensation nuclei (CN) and cloud condensation nuclei (CCN). While many observations have shown positive contributions of NPF to CCN at low supersaturation, negative NPF contributions were often simulated in polluted environments. Using the observations in a coastal city of Qingdao, Beijing and Gucheng in North China, we thoroughly evaluate the simulated number concentrations of CN and CCN using a NPF-explicit parameterization embedded in WRF-Chem model. For CN, the initial simulation shows large biases of particle number concentrations at 10–40 nm and 40–100 nm. By adjusting the process of gas-particle partitioning, including the mass accommodation coefficient of sulfuric acid, the phase changes of primary organic aerosol emissions and the condensational amount of nitric acid, the improvement of the particle growth process yields a substantially reduced overestimates of CN. Regarding CCN, SOA formed from the oxidation of semi-volatile and intermediate volatility organic vapors (SI-SOA) yield is an important contributor. At default settings, the SI-SOA yield is too high without considering the differences in precursor oxidation rates. Lowering the SI-SOA yield under linear-$H_2SO_4$ nucleation scheme results in much improved CCN simulations compared to observations. On the basis of the bias-corrected model, we find substantially positive contributions of NPF to CCN at low supersaturation (~0.2%) over the broad areas of China, primarily due to competing effects of increasing particle hygroscopicity, a result of reductions in SI-SOA amount, surpassing that of particle size decreases. The bias-corrected model is robustly applicable to other schemes, such as quadratic-$H_2SO_4$ nucleation scheme, in terms of CN and CCN, though the dependence of CCN on SI-SOA yield is diminished likely due to changes in particle composition. This study highlights the potentially much larger NPF contributions to CCN on a regional and even global basis.

## 1. Introduction

New particle formation (NPF) is a process in which gaseous vapors nucleate and form critical molecular clusters, followed by subsequent growth to larger sizes through condensation and coagulation (Kulmala et al., 2004; Kulmala et al., 2013; Lee et al., 2019). Newly formed particles could effectively grow into the size of cloud condensation nuclei (CCN) under certain supersaturation (SS), which exerts an impact on the cloud microphysical process and global radiation balance (Merikanto et al., 2007; Kerminen et al., 2018; Ren et al., 2021). In addition, the efficient nucleation and explosive growth of particles may contribute to the formation of haze (Guo et al., 2014), affecting air quality and human health (Yuan et al., 2015; Chu et al., 2019; Kulmala et al., 2021).

The overestimate of condensation nuclei (CN) in numerical models is commonly seen, despite the attempt to rectify the bias (Matsui et al., 2013; Arghavani et al., 2022). It is a common way to reduce the nucleation rate which may reduce the particle number concentration in proportion (Matsui et al., 2013). For instance, in the study of NPF in East Asia in the spring of 2009, even after lowering the nucleation rate in a regional model of WRF-Chem applied in their study, the reduced number concentration of particles at 10–130 nm remained to be overestimated (Matsui et al., 2013). Using the same regional model and a similar method to reduce the nucleation rate, Arghavani et al. (2022) found particle number concentration at 10–100 nm was still overestimated by nearly one order of magnitude, despite the effectiveness to reduce the overestimates for the smaller particles such as 2.5–10 nm. In addition to the rate of NPF, the growth process of particles also has a crucial effect on particle number concentration and size distribution. In this process, the condensation of some chemical species such as sulfuric acid, nitrate and organic gases on particles plays a major role in particle growth (Yao et al., 2018; Lee et al., 2019; Li et al., 2022), and the uncertainty of their condensation amount may lead to the bias of CN simulation.

In addition to CN, there are large discrepancies in the predicted CCN between the numerical models and observational results. Furthermore, as an important source of

CCN (Merikanto et al., 2009), the contribution of nucleation to CCN quantified by
numerical models is also highly uncertain. For example, in terms of predicting CCN,
Fanourgakis et al. (2019) evaluated the CCN concentrations simulated by 16 global
aerosol–climate and chemistry transport models with observations at 9 sites in Europe
and Japan from 2011 to 2015, and found that all models underestimated CCN
concentrations with a mean normalized mean bias of -36% at low supersaturation
(SS=0.2%). WRF-Chem models also tend to underestimate the contribution of NPF to
CCN, especially at low supersaturation. The continuous observation of CCN
concentrations throughout the year (July 2008–June 2009) carried out in Hyytiälä,
Finland, showed that under low SS, nucleation enhanced the CCN by 106% and 110%
at SS=0.1% and 0.2% respectively (Sihto et al., 2011). Observations acquired in Beijing
from July 12 to September 25, 2008, also suggested that nucleation significantly
increases CCN at all supersaturations, even when supersaturation is low (i.e., 0.07%
and 0.26%). Thus, the occurrence of NPF enhanced CCN by a factor of 1.7 and 2.2,
respectively (Yue et al., 2011).

However, previous numerical experiments behave oppositely. For instance, Matsui

et al. (2011) quantified the contribution of nucleation to CCN using WRF-chem in
Beijing in August and September 2006 and found reduced CCN under low SS, e.g.,
when SS=0.02%, the concentration of CCN is reduced by up to ~50%. They attributed
this to the fact that the small particles produced by nucleation may inhibit the growth
of the pre-existing particles (Matsui et al., 2011). Similarly, Dong et al. (2019)
conducted NPF simulations with the WRF-Chem for the summer of 2008 focusing on
the Midwest of the United States, and found that the nucleation resulted in decreased
CCN at low supersaturation (SS=0.1%). Besides, a study carried out for East Asia in
2009 also indicated that at low supersaturation (e.g. SS=0.1%), nucleation has little
impact on CCN (Matsui et al., 2013). The contrasting effects of nucleation on CCN at
low supersaturations in model and observations is not explained in these previous
studies.

At the stage of particle growth, secondary organic aerosol (SOA) formed by

atmospheric oxidation of organic vapors is a major contributor to particle growth to
CCN-related sizes (Liu and Matsui, 2022; Qiao et al., 2021). SOA formed by multi-
generational gas-phase oxidation of semi-volatile and intermediate volatility organic
compounds (S/IVOC) is called SI-SOA (Jimenez et al., 2009; Zhang et al., 2007). Zhao
et al. (2016) made a comprehensive assessment of the roles of various SOA precursors
in SOA formation in real atmosphere in China in 2010, and the results demonstrated
that evaporated POA and IVOC (i.e. S/IVOC) made a significant contribution to SOA,
contributing up to 82% to the average SOA concentration in eastern China. However,
the effect of SI-SOA on CCN has not been fully studied.
In this paper, WRF-Chem was applied to simulate the effect of the NPF on CCN
in China in February 2017. The simulated results from the WRF-Chem model are firstly
compared with observations in Qingdao, Beijing and Gucheng, exhibiting large biases
in CN. This is followed by an improvement through a few processes. At the end, the
impact of SI-SOA yield and nucleation on CCN is investigated.
**2. Data and methods**
**2.1 Observations**
The measurements used in this study were carried out over the sampling site from
February 5 to 24, 2017 at the campus of Ocean University of China (36°09′37″N,
120°29′44″E) in Qingdao, which is surrounded by residential buildings and is situated
about 10 km away from the city center. A fast mobility particle sizer (FMPS, TSI Model
3091) was applied to measure the aerosol particle size distribution for the size range of
5.6 nm to 560 nm (Liu et al., 2014b). The bulk CCN concentration is measured by a
cloud condensation nuclei counter at three different supersaturations (0.2%, 0.4% and
0.6%) and each supersaturation lasts for 20 minutes. More information about the CCN
measurement can be found in Li et al. (2015). The urban site in Beijing is located on
the roof of the building of the Chinese Academy of Meteorological Sciences (CAMS,
39°95′N, 116°33′E) in the campus of the China Meteorological Administration, close
to the main road with heavy traffic. The rural site is Gucheng (GC, 39°08′N, 115°40′E),
located in Hebei Province, surrounded by farmland, and is a representative station of
the severity of air pollution in Beijing Tianjin Hebei region. The particle number size
distribution of these two sites in the range of 4–850 nm is measured by a Tandem
Scanning Mobility Particle Sizer (TSMPS), and more information about the
instruments can be found in Shen et al. (2018).
**2.2 Model configurations**
WRF-Chem version 3.9 is used to simulate NPF events, with the main physical
and chemical parameterization settings summarized in Table 1. The spatial resolution
is 36 km by 36 km with 35 vertical layers and a model top at 50 hPa. The regional
model simulations at a higher spatial resolution may be desirable in future when urban
pollution is focused. A continuous run from February 1 to 25, 2017, was conducted,
with the first five-day results as the spin-up and discarded in the analysis.

Table 1 WRF-Chem model configurations used in this work

|  | Model configuration |
| --- | --- |
| Microphysics | Morrison 2-moment microphysics scheme (Morrison et al., 2009) |
| Planetary Boundary Layer (PBL) | YSU boundary layer scheme (Hong et al., 2006) |
| Longwave and Shortwave Radiation | RRTMG longwave and shortwave radiation |
| Land model | Unified Noah Land Surface scheme (Chen and Dudhia, 2000; Tewari et al., 2016) |
| Cumulus | Grell-3D cumulus parameterization scheme (Grell, 1993) |
| Aerosol module | MOSAIC module (Zaveri et al., 2008; Matsui et al., 2011) |
| Gas-phase Chemistry | SAPRC-99 gas-phase chemistry scheme (Carter, 2000) |


The meteorological initial and boundary conditions are driven by Climate Forecast
System model version 2 (CFSv2; (Saha et al., 2014)) reanalysis developed by National
Centre for Environmental Prediction (NCEP). The initial and boundary chemical
conditions of WRF-Chem are provided by Community Atmosphere Model with
Chemistry (CAM-Chem; (Buchholz et al., 2019)). Anthropogenic emissions for the
year of 2017 are obtained from the Multiresolution Emission Inventory for China
(MEIC, http://www.meicmodel.org/) emission dataset (Li et al., 2017; Zheng et al.,

2018).

The Model for Simulating Aerosol Interactions and Chemistry (MOSAIC) was

used to delineate dynamic gas-particle mass transfer to represent the condensation
growth of aerosol (Zaveri et al., 2008). The gas-particle partitioning of gas species on
particles is regulated by the mass transfer rate, which is related to mass accommodation
coefficient (α), a parameter involved in the model representing the probability of gas
molecules entering the bulk liquid phase (Pöschl et al., 1998). The original setting of α
for all condensing species for all size bins a in MOSAIC is 0.1 (Zaveri et al., 2008). In
the default release of WRF-Chem, MOSAIC was implemented in the sectional
framework with aerosol size distributions divided into 4 or 8 size bins spanning 39 nm
to 10 μm in diameter. To explicitly express the nucleation and the growth of newly
formed particles, the aerosol size range in the MOSAIC module was extended from 1
nm to 10 μm, with the number of aerosol size bins increased to 20 (Matsui et al., 2011;
Matsui et al., 2013; Lupascu et al., 2015; Lai et al., 2022). The calculation method of
CCN concentration in the WRF-chem model is referred to the study of Matsui et al.
(2011). Based on Köhler theory, CCN concentrations under the three given
supersaturations of 0.2%, 0.4% and 0.6% were calculated. The critical supersaturation
($S_c$) of each size bin in the WRF-chem model was calculated by the following formula:
$$S_c = \sqrt{\frac{4 \times a^3}{27 \times r^3 \times \kappa}} \tag{1}$$

$$a = \frac{2 \times \sigma}{R_v \times T \times \rho_\omega} \tag{2}$$

Where α (m) is the coefficient of the Kelvin effect, κ is the volume−averaged
hygroscopicity, calculated using these values in Table 1, r (m) is the dry diameter, σ is
droplet surface tension over water (0.076 N m$^{-1}$), $R_v$ is the gas constant for water vapor
(461.6 J K$^{-1}$kg$^{-1}$), T (K) is the air temperature, and $\rho_\omega$ is the density of water (1000 kg
m$^{-3}$).



Table 2 Hygroscopicity Parameters (κ) in the WRF-Chem Model

| Species | Hygroscopicity (κ) |
|---|---|
| Sulfate | 0.5 |
| Ammonium | 0.5 |
| Nitrate | 0.5 |
| Black carbon | $10^{-6}$ |
| Primary organic aerosol | 0.14 |
| Other inorganics | 0.14 |
| Sodium | 1.16 |
| Chloride | 1.16 |


The chemical aging process of organic aerosols (OA) is modeled by the volatility basis set (VBS) approach, which was widely used in air quality models to represent complex mixtures of thousands of organic species (Donahue et al., 2006; Shrivastava et al., 2011; Chrit et al., 2018). The VBS method classifies compounds according to the effective saturation concentration ($c^*$), which represents the proportion of the component in the gas phase to the particle phase (Donahue et al., 2006), and species with higher $c^*$ values are more volatile. The oxidation of highly volatile precursors to form relatively low volatile components represents the aging process of OA. OA consists of directly emitted primary organic aerosols and photochemically produced secondary organic aerosols (SOA) (Shrivastava et al., 2011). In this study, the simplified 2-species VBS mechanism was applied to the simulation of SOA, during which primary organic aerosol was represented by two species based on volatility with effective saturation concentration $c^*$ values (at 298 K and 1 atm) of $10^{-2}$ and $10^5$ μg m$^{-3}$ (Shrivastava et al., 2011). Primary organic aerosols with $c^*$ of $10^5$ μg m$^{-3}$ refers to S/IVOC, which is in the gas phase under most atmospheric conditions due to its high volatility, while for those primary organic matters with $c^*$ of $10^{-2}$ μg m$^{-3}$, is treated as gas phase as well in the original model. The SOA formed by photochemical oxidation of S/IVOC precursors is called SI-SOA and the SOA formed by oxidation of VOC precursors is named V-SOA. In the simplified 2-species VBS mechanism, SI-SOA ($c^*$

of $10^{-2}$ μg m$^{-3}$) is formed by the oxidation reaction of S/IVOC precursors ($c^*$ of $10^5$ μg m$^{-3}$) and OH with an oxidation rate constant of $4 \times 10^{-11}$ cm$^3$ molec$^{-1}$ s$^{-1}$. The equations for controlling the oxidation of S/IVOC precursors are as follows:

$$POA(g)_{e,c} + OH \rightarrow SI - SOA(g)_{e,c} + 0.15 SI - SOA(g)_{e,o} \tag{3}$$

$$POA(g)_{e,o} + OH \rightarrow SI - SOA(g)_{e,o} + OH \tag{4}$$

where POA(g) denotes primary organic aerosols with $c^*$ of $10^5$ μg m$^{-3}$, which reacts with OH to form SI-SOA(g) with $c^*$ of $10^{-2}$ μg m$^{-3}$. Subscripts $c$ and $o$ represent the non-oxygen and oxygen parts respectively of given species and $e$ is either the biomass or anthropogenic emission sector. In addition, SVOC and IVOC emissions corresponding to both anthropogenic and biomass burning emissions are derived based on constant emission ratio of S/IVOC to POA (Shrivastava et al., 2011). A detailed description of 2-species VBS mechanism can be found in Shrivastava et al. (2011).

**2.3 Model sensitivity formulations**

Three sets of sensitivity tests are designed and listed in Table3. The purposes of the three sets of experiments are as follows: (1) Adjust the condensation growth process of ultrafine particles in WRF-Chem model (Base, MAC, PEP, NOCD, RACD, with details in Table 3); (2) Explore the effect of SI-SOA yield on CCN (Low_Yield and High_Yield); (3) Study the effect of nucleation process on CCN under the change of SI-SOA yield (Low_Yield and High_Yield and their corresponding cases without nucleation parameterization, i.e., Low_nucoff and High_nucoff). Each scenario will be explained in conjunctions with the results.

Table 3 The sensitivity tests involved in this study

| Purposes | Simulation scenarios | Description |
|---|---|---|
| Adjust the condensation growth process of ultrafine particles | Base | Simulation with the default setting with nucleation coefficient set as $2 \times 10^{-6}\,\mathrm{s}^{-1}$, the same as Lai et al. (2022). |
| | Mass accommodation coefficient (MAC) | It is the same as Base except that the mass adjustment coefficient ($\alpha$) of gaseous sulfuric acid is adjusted from 0.1 to 0.65. |
| | POA emission phase (PEP) | It is the same as MAC except that the phase of POA is changed from gas phase to particle phase. |
| | No condensation (NOCD) | It is the same as PEP except that no $NH_4NO_3$ condenses on particles below 40 nm. |
| | Ratio method for condensation (RACD) | It is the same as PEP except that the condensation of $NH_4NO_3$ on particles below 40 nm is reduced according to the ratio of acid particles to total particles reported in Wang et al. (2014). |
| Explore the effect of SI-SOA yield on CCN (Explore the effect of nucleation process on CCN under the change of SI-SOA yield) | High_Yield | Simulation with high oxidation rate of SI-SOA formation with reaction rate constant of $5 \times 10^{-11}$ $cm^3$ $molec^{-1}$ $s^{-1}$ |
| | Low_Yield | Simulation with low oxidation rate of SI-SOA formation with reaction rate constant of $2 \times 10^{-11}$ $cm^3$ $molec^{-1}$ $s^{-1}$ |

| Explore the effect of nucleation process on CCN under the change of SI-SOA yield | High_NUCOFF | Simulations without nucleation parameterizations based on High_Yield |
| | Low_ NUCOFF | Simulations without nucleation parameterizations based on Low_Yield |

## 3. Results

### 3.1 Observational analysis

Based on the criteria (Dal Maso et al., 2005; Kulmala et al., 2012), NPF is defined as an event with the emergence of a nucleation mode with particle diameters smaller than 25 nm, lasting for 2 hours or more, followed in general by a continuous particle growth. Six NPF events were identified in February 2017 in Qingdao, on the days of 6, 9, 10, 17, 20 and 23 (Fig. 1a), yielding a frequency of ~30% and displaying a typical banana-shaped growth of particles in the particle number size distribution. Compared to a few other studies on NPF frequency in Qingdao, the results in this study are to a large extent consistent with that in the fall of 2012–2013 (30%; (Zhu et al., 2019)), slightly higher than that in summer 2016 (22%; (Zhu et al., 2019)) and lower than that in spring of 2010 (41%; (Liu et al., 2014b). The higher frequency in spring in Qingdao is consistent with the observational results at different stations in the Northern Hemisphere in Nieminen et al. (2018).

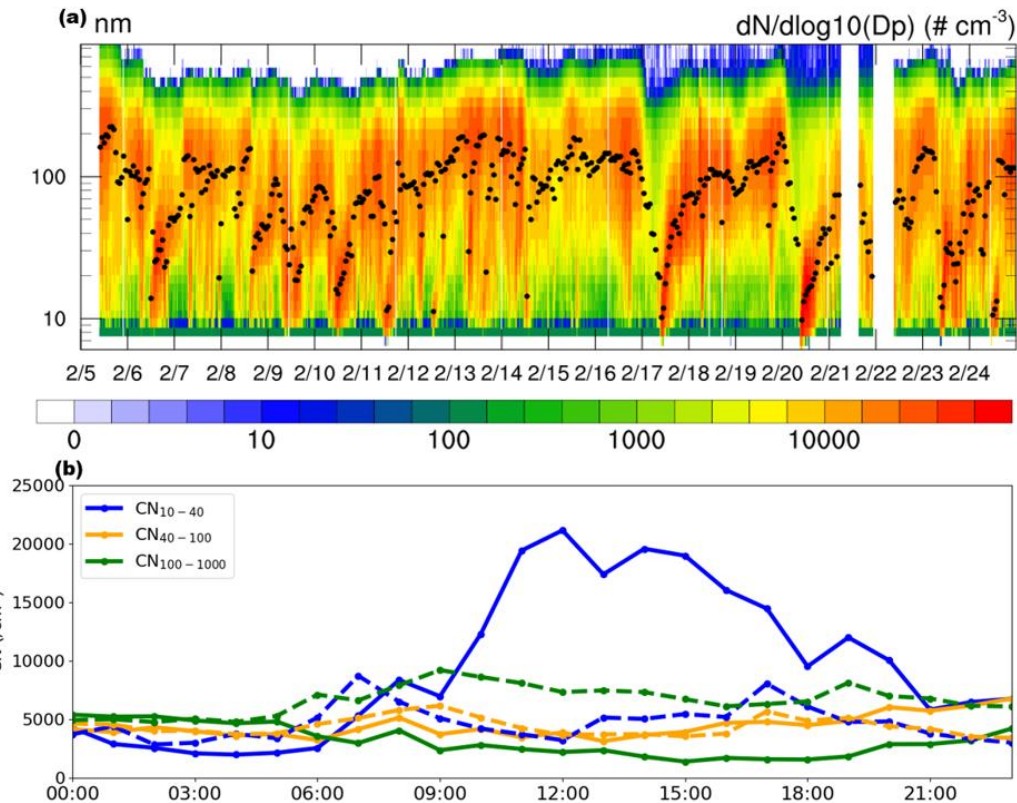

Fig. 1. Distribution of particle number concentration. (a) Temporal evolution of particle size distributions (colored shading) and geometric median diameter (GMD; dots in black) in Qingdao on February 5-24, 2017. (b) The mean diurnal variation of $CN_{10-40}$ (blue), $CN_{40-100}$ (orange) and $CN_{100-1000}$ (green) composited during the NPF (solid lines) and non-NPF (dashed lines) days on February 5-24, 2017. All times are local times (LT)

During the six NPF events identified in February in Qingdao, the mean diurnal cycle of $CN_{10-40}$ (10–40 nm) particles exhibits triple peaks (solid blue in Fig. 1b), in the morning (8:00 LT), noon (12:00–14:00 LT) and evening (19:00 LT), respectively. A comparable three-peak feature was also observed in earlier years during 2016-2018 in Qingdao (Zhu et al., 2021). The morning and evening peaks of $CN_{10-40}$, with values of ~5300 cm$^{-3}$ and ~12000 cm$^{-3}$, respectively, are likely caused by the primary emissions from traffic and cooking activities (Wu et al., 2021a; Wang et al., 2022; Cai et al., 2020). The occurrence of NPF starts approximately at 9:00 am LT, accompanied by a substantial increase in $CN_{10-40}$ compared with non-NPF days (solid vs. dashed lines, in blue), yielding a peak around noon (20000 cm$^{-3}$ during 12:00–14:00 LT). In addition,

larger particles (e.g., $CN_{40-100}$ and $CN_{100-1000}$) displayed a slow or no increase in the
afternoon.

**3.2 Model improvement in particle number concentration simulations**
Particle number concentrations, primarily in two ranges of 10–40 nm and 40–100
nm, are commonly simulated with large biases. In the smaller size range (10–40 nm),
the particle number concentration is associated with NPF and particle growth. During
NPF, despite differences among the formation mechanisms, $H_2SO_4$ is considered the
common species (Yu, 2005; Lovejoy et al., 2004), which often suffer large biases (Cai
et al., 2016; Matsui et al., 2011). In the size range of 40–100 nm, the particle number
concentration is primarily affected by the condensation growth of particles below 40
nm, which is closely related to chemical components such as SOA and nitrate. Prior to
the evaluation of particle number concentration, we first evaluate the compositions of
$PM_{2.5}$ and criteria air pollutants including $PM_{2.5}$, $PM_{10}$, $O_3$, $SO_2$, CO, and $NO_2$, showing
relatively low biases compared to observations (section S1 and Fig. S1 and Fig. S2 of
the supporting information).

**3.2.1 Bias correction of particle number concentration at 10–40 nm**
In this study, as shown in Fig. 2, comparisons of $CN_{10-40}$ between simulations (red
line in Fig. 2a) and observations (black line in Fig. 2a) results of the six NPF events
mentioned in the previous section in Qingdao in February 2017 indicate that model
overestimates $CN_{10-40}$ with mean fractional bias of 48%. As one of the major processes
affecting the particle number concentration of 10–40 nm, nucleation is governed by the
particle nucleation rate of 1 nm particles ($cm^{-3}$ $s^{-1}$), which is closely associated with the
concentration of $H_2SO_4$. For instance, in a commonly applied activation mechanism,
the nucleation rate calculated by $J^* = K_{ACT} \times [H_2SO_4]$. Note that $K_{ACT}$ is the nucleation
coefficient considering the physical properties and chemical species of nucleation
process under different environments, indicating that a lumped chemical species are
included in the scheme reflected primarily in the nucleation coefficient k, set as $2 \times 10$
$^{-6}$ $s^{-1}$ based on previous studies (Sihto et al., 2006; Riipinen et al., 2007). Dong et al.
(2019) simulated NPF occurring in the summer of 2008 in the United States using the
NPF-explicit WRF-Chem based on the activation mechanism, which overestimated the
particle number concentration at 10–63 nm by nearly doubled, even when the $K_{ACT}$
decreased by one order of magnitude (set at a very low value of $10^{-7}$ s$^{-1}$). Therefore, it
is likely that the overestimation of particle number concentration in the smaller particle
size segment is probably due to the bias of simulated sulfuric acid.

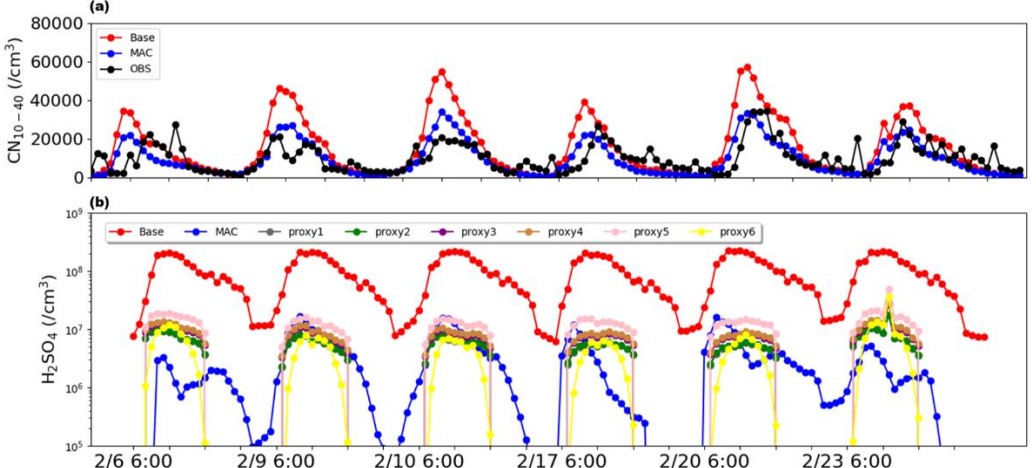


Fig. 2. Time series of (a) $CN_{10-40}$ on NPF days, where red and blue represent Base and
MAC simulation results respectively, and black represents observation results, and (b)
sulfuric acid gas concentration obtained by simulation and by proxies (dark grey: Eq.
5; green: Eq. 6; purple: Eq. 7; brown: Eq. 8; pink: Eq. 9; yellow: Eq. 10). All times are
in local times.

Measurement of sulfuric acid gases in the lower troposphere is challenging due to
the generally low ambient concentration of sulfuric acid ($10^6$–$10^7$ molecule cm$^{-3}$).
Different methods have been proposed to estimate ambient sulfuric acid concentrations
based on observations such as $SO_2$ (Petäjä et al., 2009; Lu et al., 2019; Mikkonen et al.,
2011). For instance, Petäjä et al. (2009) proposed a linear method to approximate
observed $H_2SO_4$ concentration in Hyytiälä, southern Finland. Moreover, a recent study
by Lu et al. (2019) proposed a nonlinear method to construct a number of proxies for
gaseous sulfuric acid concentration (Eq. 5–9), indicating that compared to the linear
method in Petäjä et al. (2009), the nonlinear relationship can provide more accurate
$H_2SO_4$ concentration in Beijing during February–March 2018 period. In addition, we

also used another sulfuric acid nonlinear proxy (Eq. 10) based on long-term observations in Germany, Finland, the United States, etc. (Mikkonen et al., 2011). In this study, we adopt the above six nonlinear proxy methods (referred as proxy5 to proxy10) to estimate $H_2SO_4$ in Qingdao.

$$[H_2SO_4] = 515.74 \times [SO_2]^{0.38} \times \text{Radiation}^{0.14} \times CS^{0.03} \tag{5}$$

$$[H_2SO_4] = 280.05 \cdot \text{Radiation}^{0.14}[SO_2]^{0.40} \tag{6}$$

$$[H_2SO_4] = 9.95 \times [SO_2]^{0.39} \times \text{Radiation}^{0.13} \times CS^{-0.01} \times [O_3]^{0.14} \tag{7}$$

$$[H_2SO_4] = 14.38 \times [SO_2]^{0.38} \times \text{Radiation}^{0.13} \times [O_3]^{0.14} \tag{8}$$

$$[H_2SO_4] = 0.0013 \times [SO_2]^{0.38} \times \text{Radiation}^{0.13} \times CS^{-0.17} \times ([O_3]^{0.14} + [NO_x]^{0.41}) \tag{9}$$

$$[H_2SO_4] = 8.21 \times 10^{-3} \times [SO_2]^{0.62} \times \text{Radiation} \times (CS \times RH)^{-0.13} \tag{10}$$

where $[SO_2]$, $[O_3]$ and $[NO_x]$ (molecule cm$^{-3}$) represents concentration of observed $SO_2$, $O_3$ and $NO_x$, respectively. "Radiation" (W m$^{-2}$) is global radiation. RH (%) is the relative humidity, and CS (s$^{-1}$) is the condensation sink, which is calculated based on observed particle distribution.

The simulated $H_2SO_4$ concentration from the Base simulation (dots in Fig. 2b) is compared with observations obtained by proxies (see Fig. 2b), indicating that Base simulations apparently overestimate by one order of magnitude compared to the $H_2SO_4$ estimated by proxies. The overestimation has been frequently reported previously, i.e., over Beijing (Matsui et al., 2011), which ascribes the bias to the overestimation of the $SO_2$ concentration. In a more recent study, the sensitivity of $H_2SO_4$ to $SO_2$ is tested, and the result shows that even when $SO_2$ is reduced to an unrealistically low level, the simulated $H_2SO_4$ is still more than one order of magnitude higher than the observed value (Lai et al., 2022), suggesting that the $SO_2$ concentration cannot fully explain the overestimates.

In addition to the precursor of $H_2SO_4$, the mass accommodation coefficient (α), representing the probability of impaction of a gaseous molecule on a liquid surface and entering the bulk liquid phase, is another important factor affecting the concentration of sulfuric acid gas. In the public release of WRF-Chem, mass accommodation

coefficient is typically set to a low value of 0.1 for all gas species under different
volatility during the condensation process, including $H_2SO_4$ (Davidovits et al., 2004;
Zaveri et al., 2008). Recent studies indicate that the low mass accommodation
coefficient value may not be applicable to the low volatile gases, which tend to have a
mean mass accommodation coefficient value of 0.7 and close to the unity (Krechmer et
al., 2017). In fact, an earlier study has indicated based on experimental determination,
the mass accommodation coefficient of $H_2SO_4$ vapor in sulfuric acid aqueous solution
was measured, and the best fit value was 0.65. Accordingly, a sensitivity simulation was
conducted by adjusting the mass accommodation coefficient of $H_2SO_4$ from 0.1 to 0.65,
referred to as MAC.

This simulation brought the $H_2SO_4$ concentration (see Fig. 2b) much closer to the

calculated results from proxies, and the corresponding biases reduced by approximately
an order of magnitude. Notably, the MAC simulation decreases the overestimate of
sulfuric acid gas concentration, resulting in a lower particle formation rate. The MAC
simulation also significantly reduces overestimate of $CN_{10-40}$ (Fig. 2a), and mean
fractional bias compared to observations decreases from 48% to 1%.

**3.2.2 Improvement of particle number concentration simulations at 40–100 nm**

The number concentration of particles in the 40–100 nm range is mainly affected

by the coagulation and condensation processes. While the coagulation process tends to
largely affect ultrafine particles below 10 nm than those with larger sizes (Wu et al.,
2011), the condensation growth of particles during gas-particle partitioning at sizes of
10–40 nm, to a large extent, governs the variations in number concentration of 40–100
nm particles. The condensation process is primarily controlled by gas-particle
partitioning of chemical species, which may change the chemical composition of
particles, such as organic compounds and inorganics including sulfate, nitrate and
ammonium.

Among the species contributing to the condensation growth of particles at 10–40

nm, the organic compounds with $c^*$ of $10^{-2}$ $\mu g\ m^{-3}$ play the dominant role (Pierce et al.,
2011). In the current model setting, the low volatile organic matter of $10^{-2}$ $\mu g\ m^{-3}$ comes

from two gas-phase sources, including the direct emission of primary organic aerosol (POA) and SOA formed from S/IVOC (SI-SOA), conducive to condensation on particles. While the condensation of gaseous SOA is in general reasonable, the gas phase emissions of POA may be problematic. For instance, previous studies suggested that POA is in gas phase close to the emissions source. However, with rapid dilution and cooling in the atmosphere away from the source, most POA condenses to particle-phase (Roldin et al., 2011a; Roldin et al., 2011b; Shrivastava et al., 2008). Therefore, away from the emissions source POA, being in the particle phase, will not be involved in the growth of newly formed particles. Therefore, POA may not contribute to particle growth away from the emission sources, which caused different size distributions of POA compared to when it was emitted in the gas-phase (Fig. S3a vs. Fig. S3b). Emitting low volatility POA in the particle phase eliminates the unreasonable quasi-banana shape pattern exhibiting concomitant growth of newly formed particles with increasing mass concentration of POA.

The composition analysis (Fig. S3c) in the 10–40 nm particles mass from the model results indicates that organic compounds mentioned above only account for 21% of total mass (sulfates, nitrates, ammonium salts and organics) in this size range and the dominant species is nitrate which accounts for 51% of total mass, exhibiting inconsistencies with the previous studies which in general indicates a much smaller contribution of nitrate. For instance, Liu et al. (2014a) suggested that over North China Plain in summer 2009, organic matter accounted for 77% of particles around 30 nm, while the sum of $SO_4^{2-}$, $NO_3^-$ and $NH_4^+$ only accounted for 18%. Recent observations conducted in Beijing also indicated that particles at 8–40 nm are mainly composed of organic matter (with mass fraction of ~80%) and sulfate (with mass fraction of ~13%), while nitrate content is very low (with mass fraction of ~3%) (Li et al., 2022). Another study showed that nitrate accounted for 7–8% at urban sites and 17% at rural sites for particles mass in 7–30 nm in the United States in 2007 (Bzdek et al., 2012). Therefore, the potentially too high modeled nitrate fraction in 10–40 nm in this study is tightly associated with the condensation process, with the specific reasons explained below.

The condensation of nitric acid on particles is highly constrained by the particle

acidity. The acidity in smaller particles (i.e., 10–40 nm) tends to be higher than that in large particles, primarily due to the larger condensation of $H_2SO_4$ (Lu et al., 2022), and particles with sizes greater than 40 nm have a much weaker acidity or are nearly neutral. For example, observed evidence has shown that acidic ultrafine particles account for a large proportion of ultrafine particles from 22 December 2010 to 15 January 2011 in Hong Kong, e.g., 65% for particles within 5.5–30 nm (Wang et al., 2014).

In the model, a particle is determined to be in solid phase when the ambient relative humidity is lower than the mutual deliquescence relative humidity of the particles (Zaveri et al., 2005; Zaveri et al., 2008), which is in general suitable for particles dominated by inorganics. In the study area, the results indicate that at most conditions relative humidity are relatively low and the particles are in solid phase, in which the condensation process is not affected by particle acidity and the condensation of nitric acid on particles is directly calculated based on the gas-particle equilibrium concentration (Zaveri et al., 2008). However, for particles below 40 nm, the main compositions are likely to be organic matter (Zhu et al., 2014; Ehn et al., 2014), which tends to be in liquid phase (Virtanen et al., 2011; Cheng et al., 2015), under which the condensation of nitric acid is strongly constrained by acidity. Therefore, the phase misrepresentation ignores the weakening effect of acidity on nitric acid condensation, resulting in too high nitrate therein.

To overcome this issue, we propose a ratio method for condensation (RACD) to partition the condensation of nitric acid on particles under 40 nm, by applying a ratio of the number concentration of non-acidic particles to ultrafine particles. The method is based on two assumptions, including: 1) little condensation of nitric acid on particles with strong acidity (Lu et al., 2022); 2) the condensation of nitric acid on particles is proportional to the ratio of the number concentration of non-acidic ultrafine particles to the total particles, despite the existence of uncertainties. Fig. S4 depicts the average particle number concentration and acid particle in the 1 to 40 nm range, calculated based on Wang et al. (2014). The ratio of non-acidic particles is 8% for particles below 10 nm, 18% for particles at 10–15.8 nm, 30% for particles at 15.8–25.1nm, and 55% for particles at 25.1–39.8 nm (Fig. S4). Note that the ratio is based on measurements

acquired at a single site in Hong Kong, therefore more observational studies are needed
to warrant the robustness of the method. Alternatively, the condensation of nitric acid
on particles in bins from 1 nm to 40 nm is completely suppressed, referred to as NOCD.

The simulation results based on the two methods (RACD and NOCD) are shown in

Fig. 3. Compared to MAC, RACD simulations reduce previously noted overestimation
of particle number concentration in the 40–100 nm size range (Fig. 3b), with the mean
fractional bias decreases from 83% to 63%. In addition to the amount of nitrate
condensation during particle growth mentioned above, the overestimation of particle
number concentrations in the 40–100 nm range may be attributed to nucleation process.
More specifically, in the $H_2SO_4$-$H_2O$ binary nucleation mechanism used in this study,
when the concentration of sulfuric acid gas is reduced (Section 3.2.1), the resulting
decrease in nucleation rate leads to a slight decrease in particle number concentration
at 40–100 nm relative to Base (mean fractional bias from 98% to 83%). Apart from that,
it may also be related to the choice of nucleation parameterization scheme. For example,
using a global chemical transport model GEOS-Chem with a nucleation mechanism in
which formation rate is a function of the concentrations of sulfuric acid and low-
volatility organics, Yu et al. (2015) overestimated the concentration of particles in the
10–100 nm range by 161% at nine sites in the summer in North America. A possible
explanation for this overestimation was given by the uncertainty of the predicted
concentration of organic compounds involved in organics-mediated nucleation
parameterization. After they switched to another scheme of the ion-mediated nucleation
mechanism without organic matter, the number becomes 27% lower than the
observations (Yu et al., 2015). The test based on different schemes is beyond the scope
of the study, which is therefore not investigated.

Moreover, the overestimation of particles over 100 nm ($CN_{100–1000}$; Fig. 3c), which

have a strong influence on CCN, also decrease in the RACD simulation. Thus, the mean
fractional bias decreases from 25% (MAC) to 1%. Note that the slight increase of $CN_{10–40}$
through the application of RACD, can be linked to the decrease of nitrate
condensation, and leads to weakened particle growth and enhanced particle number
concentration at 10–40 nm (Fig. 3a). The alternative method by completely removing

the nitrate condensation (NOCD) yields even better performance in particle number concentration of 40–100 nm (mean fractional bias of 34%), indicating the feasibility by reducing the nitrate condensation. The proportion of nitrate simulated by RACD is 23%, closer to values reported in past observations (Bzdek et al., 2011; Bzdek et al., 2012), while the nitrate (1%) in the scenario of NOCD seems to be too low. Considering the limited observational information obtained based on previous studies, RACD is applied in this study.

In addition to Qingdao, we evaluate the model performance over a few other sites, including one site over urban Beijing and the other one over the rural area of Gucheng, yielding consistent improvements in model simulations (Section S2; Fig. S5-S7). Moreover, we select another empirical scheme, e.g., kinetics, and one classical nucleation scheme, indicating the empirical scheme of activation scheme is in general a good option in this study (Section S2; Fig. S8-S10; Table S1-3).

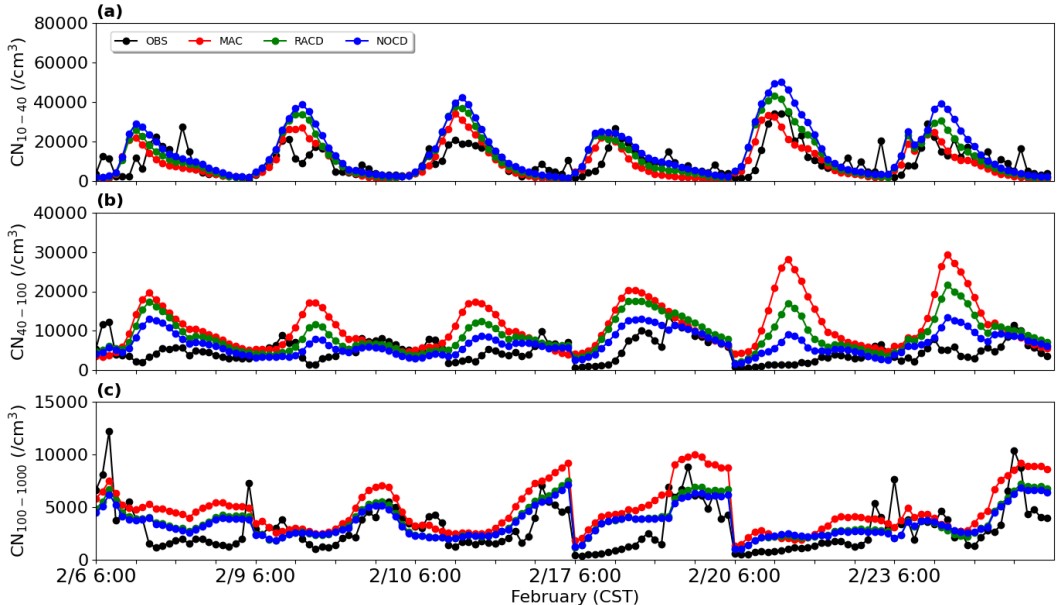

Fig. 3. The time series of (a) $CN_{10-40}$, (b) $CN_{40-100}$ and (c) $CN_{100-1000}$ on NPF days in Qingdao on February 5-24 simulated from MAC (marked in red), NOCD (marked in blue) and RACD (marked in green) as well as from observations (OBS) (marked in black). All times are local times.

**3.3 Substantial contributions of SI-SOA to CCN**

Compared with the original model setting, after adjusting the growth process of ultrafine particles (RACD), the number concentration of particles tends to decrease, especially for particles above 40 nm. Ultrafine particles above 40 nm are important sources of CCN (Dusek et al., 2006), in this way, the number concentration of CCN also tends to decline. In addition, in the Base case, we found that the model overestimated $CCN_{0.4\%}$ and $CCN_{0.6\%}$, with mean fractional bias being 64% and 87%, respectively. After adjusting the condensation growth process of ultrafine particles, under high supersaturation (i.e., $CCN_{0.4\%}$ and $CCN_{0.6\%}$), the capability of the model in reproducing the CCN is improved. RACD reduces the overestimation of $CCN_{0.4\%}$ and $CCN_{0.6\%}$, with mean fractional bias reduced to 30% and 56%, respectively, although the overestimates still exist (Fig. S11b, c). However, for low supersaturation (i.e., $CCN_{0.2\%}$), the decrease of number concentration of CCN is too large, and mean fractional bias decreases from 7% to -45% (Fig. S11a), therefore the bias will be further adjusted later.

In addition to the growth process, the remaining overestimate of CCN under high SS and underestimate of CCN over low SS is likely to be influenced by the chemical compositions involved in the activation of ultrafine particles into CCN. Specifically, ultrafine particles can grow up to CCN size under certain SS (Pierce and Adams, 2007). This process is influenced by both particle size and hygroscopicity, and hygroscopicity is closely related to the chemical composition of particles (Petters and Kreidenweis, 2007). In particular, inorganic compounds generally increase particle hygroscopicity, increasing CCN. SOA has dual effects on CCN since it decreases particle hygroscopicity but also promotes growth of particles, and these two effects are competitive with each other (Wu et al., 2015; Zaveri et al., 2021). Ultrafine particles must grow to a critical size to be activated into CCN (Dusek et al., 2006). SOA act as a major contributor in promoting the condensational growth of ultrafine particles to the critical size, facilitating particles activation into CCN. In contrast, SOA tends to reduce the hygroscopicity of particles, leading to a diminished ability of activation to CCN (Wu et al., 2015). These two competing effects work together and modulate the number

of CCN. Moreover, considering that SI-SOA is the main SOA component on ultrafine
particles (Fig. S11d), the effect of SI-SOA on CCN is therefore explored in this study.
Considering SI-SOA is a product of S/IVOC oxidation, the oxidation rate of
S/IVOC is tightly associated with CCN, which likely affects the bias of CCN. In the
original model setup, the oxidation rate is set to be a constant of $4 \times 10^{-11}$ $cm^3$ $molec^{-1}$
$s^{-1}$ for all S/IVOC. However, a recent study (Wu et al., 2021b) proposed that the
oxidation rate can be as high as $5 \times 10^{-11}$ $cm^3$ $molec^{-1}$ $s^{-1}$ such as for polycyclic aromatic
hydrocarbons (PAHs), close to the original model value, but can be as low as half (i.e.,
$2 \times 10^{-11}$ $cm^3$ $molec^{-1}$ $s^{-1}$) of the original modeling setting for S/IVOC species except
PAHs (O-S/IVOCs). It is noteworthy that the oxidation rates of $5 \times 10^{-11}$ and $2 \times 10^{-11}$
in general represent the upper and lower bounds (Zhao et al., 2016; Wu et al., 2021b).
To delve into how oxidation rate affects CCN, we set up a few numerical
experiments (Table 3) to investigate the response of CCN to the oxidation rate of
S/IVOC at three supersaturations (0.6%, 0.4%, 0.2%), including cases of High_Yield
and Low_Yield. As it is shown in Fig. 4, decreasing the oxidation rate (Low_Yield)
leads to a reduction of ~10% of CCN at high supersaturation (i.e., CCN0.6%) as
compared to the High_Yield simulation. This behaviour is a consequence of the
decrease of particle number concentrations associated with Low_Yield, particular of
the particles close to the critical diameter (40–100 nm). In this case, the effect of particle
size dominates the hygroscopicity. In contrast, at a lower supersaturation (CCN$_{0.2\%}$),
CCN increases by 42% when the oxidation rate is switched from a high to a low value,
which is due to the smaller fraction of SI-SOA contributing to particulate mass when
the oxidation rate is low. In this case, relative to SOA, a larger fraction of other particle
constituents such as inorganics, increase the volume weighted particle hygroscopicity
(Dusek et al., 2006) which causes the increase of CCN number. This means that the
effect of hygroscopicity on CCN surpasses the influence on particle size at low
supersaturations. This conclusion is consistent with the observation conducted by Ma
et al. (2016) in the North China Plain in 2013, which suggested that along with the
decrease of SS, the particles that can be activated to CCN is more sensitive to changes
of particle hygroscopicity. Similarly, based on observational data in northern China in
summer, Wang et al. (2023) found that CN in 2020 is lower than that in 2014 due to
particulate pollution control, however, the particles become more easily activated,
attributable to the larger extent of decrease in organic matters compared to inorganics,
leading to enhanced particle hygroscopicity and more conducive to activation.

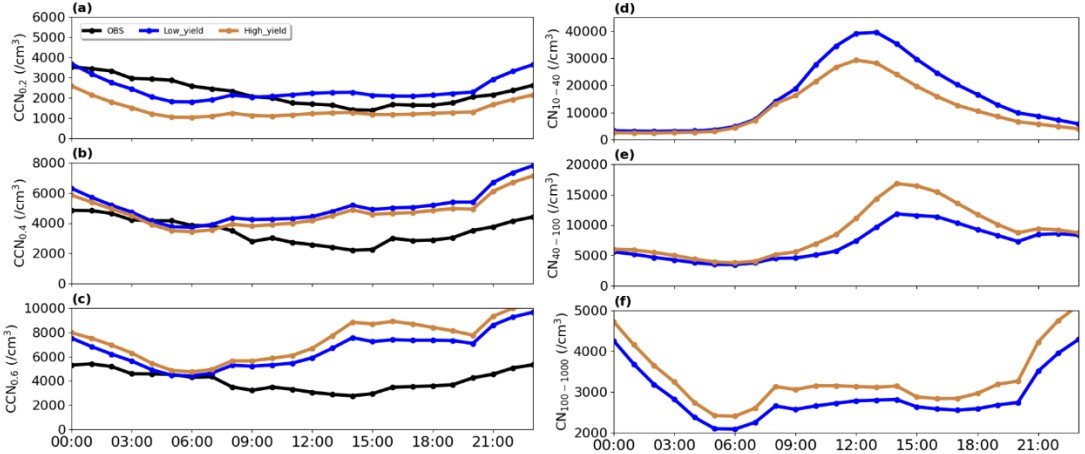


Fig. 4. Average diurnal variation of (a) $CCN_{0.2\%}$, (b) $CCN_{0.4\%}$ and (c) $CCN_{0.6\%}$ and (d)
$CN_{10–40}$, (e) $CN_{40–100}$, (f) $CN_{100–1000}$ on NPF days in Qingdao on February 5-24, 2017,
in Low_yield and High_yield simulations, shown as blue and brown lines, and black
lines represent observation results.

Furthermore, compared to the high yield of SI-SOA, the low SI-SOA yield results
in a high CCN concentration under low SS and low CCN concentration under high SS.
Therefore, both the underestimates of $CCN_{0.2\%}$ (mean fractional bias of -45%) and
overestimates of $CCN_{0.6\%}$ (mean fractional bias of 56%) mentioned above are improved,
with mean fractional bias of $CCN_{0.2\%}$ and $CCN_{0.6\%}$ reaching 7% and 43%, respectively
(Fig. 4a, c). This result suggests that the oxidation rate of S/IVOC is possibly closer to
the low value, which is understandable based on Wu et al. (2021b), who found that the
amount of O-S/IVOCs, which corresponds to a low oxidation rate, is in general much
larger (i.e., 20 times) than that of PAHs with a high oxidation rate.
In addition to the single site of Qingdao, we further explore the impact of SI-SOA
yield on CCN from a larger spatial coverage (Fig. 5). Consistent with the mechanism
revealed over Qingdao, even from a larger spatial perspective, a lower oxidation rate of

S/IVOC essentially enhances CCN at a lower SS (e.g., $CCN_{0.2\%}$; Fig. 5a) with the highest increase over North China Plain area (Fig. 5a), and weakens CCN (i.e., by 10–20% over Beijing-Tianjin-Hebei) at a higher SS (Fig. 5c), particularly over the dense emission area (Fig. S12). It is worth noting that in the 2-species VBS mechanism used in our study, all S/IVOC in the inventory is calculated based on a constant emission ratio of S/IVOC to POA from all source categories (Shrivastava et al., 2011), which may miss part of S/IVOC due to different emission ratios of POA from different source (Chang et al., 2022). In addition, the simplified VBS mechanism used in our study does not take into account the multi-step oxidation of organic species, which may introduce some uncertainties. To be more specific, in the 2-species VBS mechanism, SI-SOA with effective saturation concentrations ($c^*$) of $10^{-2}$ μg m$^{-3}$ is formed by the vapor phase oxidation of S/IVOC vapors with $c^*$ of $10^5$ μg m$^{-3}$, reducing volatility by 7 orders of magnitude. The process of one-step oxidation does not mean to represent a physical process, but to parameterize the mean effect of a complex process of SOA formation (Shrivastava et al., 2011). However, in the real atmosphere, the gaseous VOCs often undergo multi-generational oxidation to form SOA (Garmash et al., 2020), during which the properties and composition of SOA change substantially. For instance, by adding the formation chemistry associated with multi-generational oxidation, Zhao et al. (2020) found improved simulations of vertical aerosol profile in the Amazon free troposphere compared to the simplified VBS mechanism.

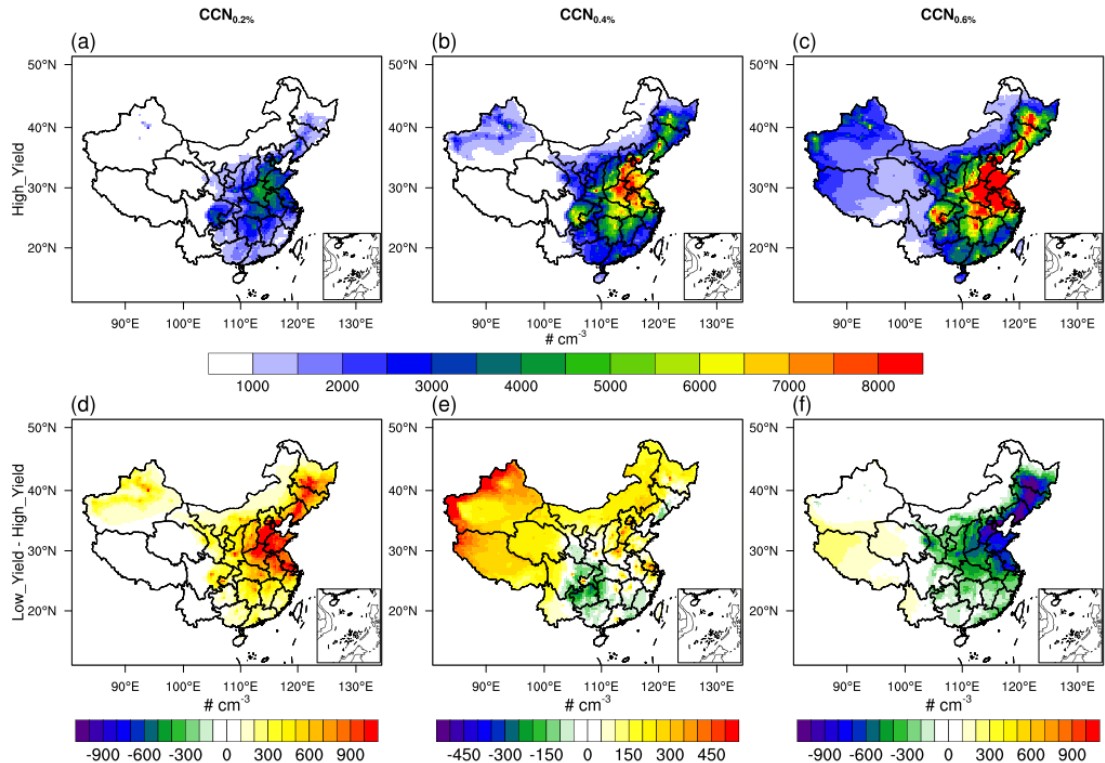

Fig. 5. Spatial distributions of CCN concentrations at different supersaturations (SS), (a) and (d) are $CCN_{0.2\%}$, (b) and (e) are $CCN_{0.4\%}$, and (c) and (f) are $CCN_{0.6\%}$. The top panels exhibit the results from the High_Yield simulation, and the bottom panels shows the difference between the Low_Yield and High_Yield simulations.

**3.4 Contribution of nucleation to CCN under different SI-SOA yields**

Considering the importance of nucleated particles on CCN (Yu et al., 2020; Westervelt et al., 2013), we further investigate the influence of nucleation on CCN under different SI-SOA yield conditions discussed above.

As shown in Fig. 6, in simulations close to the original model setting (High_Yield), when SS is low (i.e., SS=0.2%), the nucleation process tends to reduce the CCN by ~10–50%. In contrast, when the SS is high (0.6%), the nucleation results in a significant increase in CCN in most regions of China. When the yield of SI-SOA is adjusted to a lower level, the nucleation process has a positive contribution to CCN under both low and high SS. Especially, when SS is low (0.2%), the sign reversal, i.e., from negative (Fig. 6a) to positive (Fig. 6d) contributions of NPF to CCN along with the decrease of SI-SOA yield, i.e., the increase is concentrated in the eastern China with an average of

10–20%. The primary mechanism lies in that along with the decrease of SI-SOA yield,
the smaller fraction of SI-SOA yields an increase in hygroscopicity, which surpasses
the suppression effect on particle growth due to reduced SI-SOA formation. In the real
atmosphere, when the supersaturation is usually low, e.g. about ~0.1% in polluted areas
(Kalkavouras et al., 2019; Hudson and Noble, 2014), CCN will likely reduce with
increasing oxidation rate of S/IVOC and corresponding SI-SOA formation.

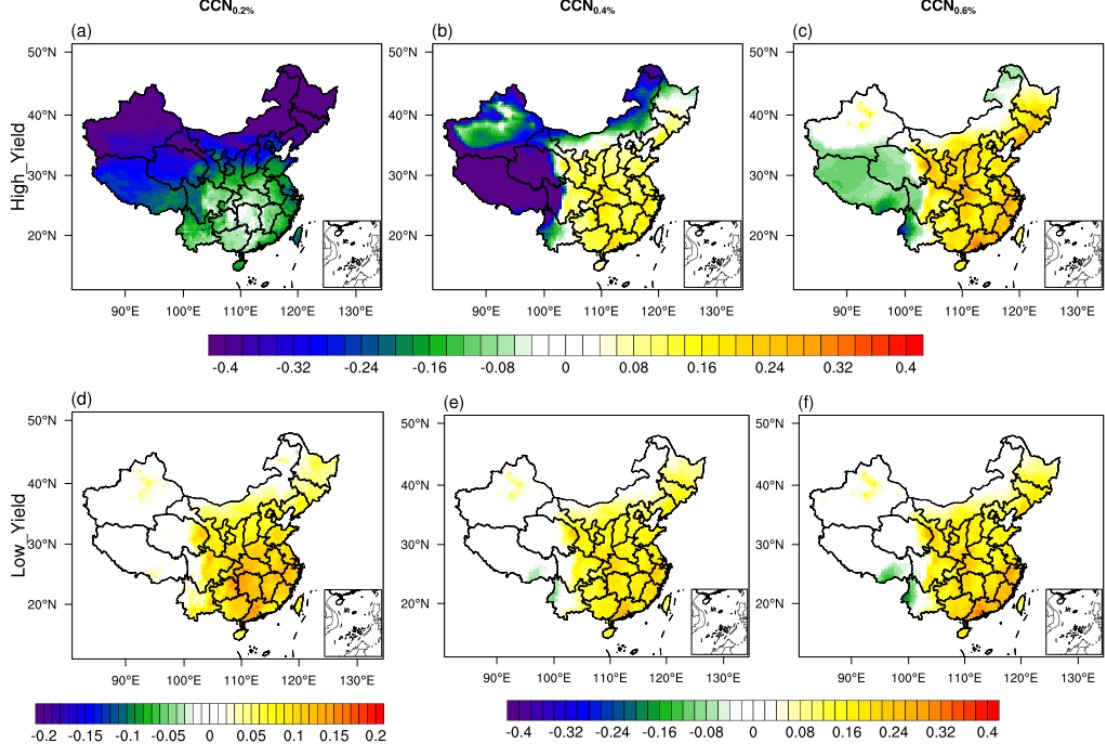


Fig. 6. Spatial distribution of contribution of nucleation to CCN calculated by the ratio
of the difference between the parameterization with and without nucleation to the
parameterization with nucleation under different SI-SOA yields in China in February
2017. (a), (d) is $CCN_{0.2\%}$, (b), (e) is $CCN_{0.4\%}$, (c), (f) is $CCN_{0.6\%}$. The upper panel and
lower panel represent High_Yield and Low_Yield simulation respectively

In addition to the linear-$H_2SO_4$ nucleation mechanism, one more empirical scheme

of kinetics nucleation is selected, which assumes that the nucleation rate is proportional
to the square of the concentration of sulfuric acid ($J = K[H_2SO_4]^2$), to investigate the
effect of nucleation on CCN. Substantially positive contributions of nucleation to CCN
is found when the low SI-SOA yield is applied, consistent with what was shown based
on the linear-$H_2SO_4$ nucleation scheme (Fig. S13). However, nucleation contributes
positively to CCN even when the SI-SOA yield is high in the quadratic-$H_2SO_4$
nucleation scheme (e.g., kinetics nucleation scheme). When more sulfuric acid
molecules participate in nucleation under this scheme than the linear-$H_2SO_4$ nucleation
scheme, the particles are more easily hygroscopically activated to CCN, which is
equivalent to the effect of a reduction in organic components in the linear-$H_2SO_4$
nucleation scheme (e.g., activation-type nucleation scheme). The results from this study
show the importance of assessing the simulated effects of the nucleation scheme on not
only the formation and growth process of particles but also climate factors such as CCN
using observations.

**Conclusions and discussions**
In this study, WRF-Chem explicit-NPF simulations, with linear-$H_2SO_4$ nucleation
scheme (e.g., activation-type nucleation scheme), are used to investigate the observed
wintertime NPF events and their contribution to CCN in China. Based on observations
in a typical coastal city of Qingdao, as well as in the cities of Beijing and Gucheng over
North China Plain, we identify high biases of the model simulated CN and CCN
concentrations. Therefore, we updated and improved the parameterization setting on
particle growth in the model, mainly including: (1) adjusting the mass accommodation
coefficient ($\alpha$) to from the default value of 0.1 to 0.65, an important parameter for
sulfuric acid condensation; (2) proportionally reducing the condensation amount of
nitric acid on particles below 40 nm; (3) changing the emitted low-volatility POA from
gas to particle. Through these adjustments, the capability of the model in reproducing
CN and CCN is substantially improved, leading to better agreement with the observed
results, which significantly reduces the overestimation of $CN_{10-40}$ (mean fractional bias
decreases from 48% to 1%) and $CN_{40-100}$ (mean fractional bias decreases from 98% to

63%).


For CCN, due to the crucial role of SI-SOA in promoting the growth of ultrafine particles, on the basis of previous studies, we lower the oxidation rate of S/IVOC and hence the production rate of SI-SOA, which weakens the growth of particles to reach the critical size of CCN activation, but enhances particulate hygroscopicity favoring the activation to CCN. When the yield of SI-SOA is adjusted to the lower bound of literature value, $CCN_{0.6\%}$ is reduced by ~10% and is closer to observations. At low SS ($CCN_{0.2\%}$), the decrease of SI-SOA yield has greater effects on the increase of particle hygroscopicity compared to the effect of the reduction of particle size due to the decrease of condensation growth. It results in an increase of CCN (as large as ~42%) in better agreement with observations. Under low SS conditions, common in the atmosphere, a 2.5-fold reduction in SI-SOA yield results in a substantial increase of CCN that switches from a negative contribution of new particle formation to CCN from -50%~-10% to a positive contribution of 10~20%.

In addition to activation nucleation scheme, we have also tested a few other schemes such as the quadratic-$H_2SO_4$ nucleation scheme (e.g., kinetics nucleation scheme). Under this scheme, the bias-corrected method abovementioned is applicable to improving the simulations of concentrations of CN and CCN. It is noteworthy that the dependence of CCN on the SI-SOA yield is diminished, showing that under both high and low yields of SI-SOA, there are positive contributions of NPF to CCN. This is likely due to the increase in the amount of sulfuric acid involved in nucleation, making it more hygroscopic and easier to activate to CCN, and the high content of inorganic species makes them less sensitive to changes in SI-SOA yield, which deserves further investigation.

**Competing interests.** At least one of the (co-)authors is a member of the editorial board of Atmospheric Chemistry and Physics.

**Acknowledgements.** This research was supported by grants from the National Natural Science Foundation of China (42122039) and Fundamental Research Funds for the Central Universities (202072001). Y.W. was supported by the National Science

Foundation Atmospheric Chemistry Program. M.S. was supported by the U.S.
Department of Energy (DOE) Office of Science, Office of Biological and
Environmental Research (BER) through the Early Career Research Program and the
Atmospheric System Research (ASR) program.

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
