# Peer review of "Substantially positive contributions of new particle formation to Cloud Condensation"

_EGUsphere, 2023_

## Referee Comment (RC1)

This study improves the simulation accuracy of particle and CCN number concentration through adjusting serval parameters such as the mass accommodation coefficient. The results are interesting and has a good potential to improve the model of NPF and its impact on CCN number concentration. This study is within the scope of the journal ACP. I recommend this paper for publication after the following issues are resolved.

A major concern is that this study is based on the comparison of measurement and model data at only one costal site. However, aerosol and CCN data has a large difference between the costal and inland sites. It is expected that the concentrations of aerosol and CCN is higher at inland sites than those in costal sites. Therefore, the Base overestimation of CCN may not exist or is weak in inlands. There are many measurements of aerosol and CCN in China. I suggest that the authors can do this work based on more measurement data, especially the data from polluted regions.

Line 147. Li et al (2015) shows the measurement of bulk CCN, not the measurement of size-resolved CCN. The authors should give more information about the size-resolved CCN measurement, including the flow set, multi-charge calibration method, data quality control and so on.

Line 262: The extremely low volatility volatile organic compounds (ELVOCs) can also have a contribution for the nucleation events.

Line 458-460: The activation critical particle size at SS=0.2% is about 120~130 nm. It is interesting that the $CCN_{0.2\%}$ has an obvious underestimation when $CN_{10-1000nm}$ has an overestimation (Figure 3c) for the RACD scheme. The change of activation ratio ($N_{CCN}/N_{CN}$) can be further analyzed.

Lie471-473: Why secondary inorganic aerosols (SIA) is not the major contributor of condensational growth?

Figure 5. The authors emphasize that the model modification has a largest impact on the simulation of CCN over the North China Plain. It is necessary to verify the result based on the measurement in this region.

Technical suggestions:
Line 249: "particles growth" should be "particle growth"
Line 269: "NFP" should be "NPF"
Line 316-328: This paragraph use "$a$" and "MAC" as the abbreviation of "mass accommodation coefficient". I suggest using a unified symbol in text.

---

## Referee Comment (RC2)

The study improved the simulation of aerosol formation processes in the NPF-explicit WRF/Chem model. The amount of work done was impressive. The results are meaningful to aerosol modeling. Overall, the manuscript is well written. However, some details are missing and some expressions need to be revised. I recommend the publication of this work after the following concerns are addressed.

Major comments:

(1) As stated by the authors, new particle formation (NPF) and subsequent particle growth are important sources of condensation nuclei (CN) and cloud condensation nuclei (CCN). NPF is an objective phenomenon and therefore its contribution to CN and CCN is definitely positive. "The nucleation resulted in decreased CCN" is only based on the simulation analysis. The authors should be cautious about the discussions relevant to "negative contribution".

(2) When using empirical formulation to calculate nucleation, the overestimation of CN could not be simply attributed to the high prefactor/coefficient in formulation. The empirical formulation has physical flaws because it could not consider the influence of other factors besides sulfuric acid, e.g., temperature and condensation sink.

(3) The model bias in this study is only based on the comparison of simulation against measurements at one costal site. Some discussions may be not applicable to other regions. I suggest that the authors can do some work based on more measurements if possible. At least, discussions on existing modeling studies in China or the influence of factors affecting model performance in other regions of China should be added.

(4) The comparison of the simulated mass concentration of aerosol components and gaseous precursors against observations is encouraged to present before evaluating the model performance in CN.

(5) The method of calculating CCN is necessary in Sect.2.

(6) How is the emission of IVOC considered?

(7) Is Knudsen effect considered when calculating the condensation of gases onto

nano-size particles?

(8) The simplified SOA simulation in this study did not consider the multi-stage oxidation of organic species, which can affect the microphysical properties of organic species in particle growth processes. Discussions on this issue is suggested.

(9) The model resolution is not high enough to well resolve the urban pollution. In discussing the model performance, this factor should be taken into account.

(10) Fig.3 shows a different temporal variation of $CN_{10\text{-}40}$ from that of $CN_{40\text{-}100}$ which shares a similar pattern with that of $CN_{100\text{-}1000}$. Primary emission may be a dominant factor contributing to model bias in $CN_{40\text{-}100}$. Is the diurnal variation of emission considered in emission data? Correlation coefficient between the simulation results and the observations is recommended to show.

Technical suggestions:

L191 and L193, organic aerosols -> organic matters

L269, NFP-explicit -> NPF-explicit

In Fig.2 and Fig.3, horizontal label bar is better

Higher and narrower Fig.4 with horizontal label bar is better.

In fig.5, a unified color bar for fig.5a, b, and c would be better.

The following papers are for reference:

Wang et al., 2015. Connection of organics to atmospheric new particle formation and growth at an urban site of Beijing. Atmospheric Environment.

Zhu et al., 2022. Airborne particle number concentrations in China: A critical review. Science of the Total Environment.

Chen et al., 2021. Global-regional nested simulation of particle number concentration by combing microphysical processes with an evolving organic aerosol module, Atmospheric Chemistry and Physics.

Yu et al., 2010. Spatial distributions of particle number concentrations in the global troposphere: Simulations, observations, and implications for nucleation mechanisms, Journal of Geophysical Research.

Spracklen et al., 2008. Contribution of particle formation to global cloud condensation nuclei concentrations. Geophysical Research Letters.

---

## Author Comment (AC1)

**Response to Reviewer #1** (our response in **colour**)

We thank the reviewer for the constructive comments to help us further improve the manuscript. Please see the detailed responses to your comments below.

**Reviewer #1:**

The study improved the simulation of aerosol formation processes in the NPF-explicit WRF/Chem model. The amount of work done was impressive. The results are meaningful to aerosol modeling. Overall, the manuscript is well written. However, some details are missing and some expressions need to be revised. I recommend the publication of this work after the following concerns are addressed.

**Major comments:**

(1) As stated by the authors, new particle formation (NPF) and subsequent particle growth are important sources of condensation nuclei (CN) and cloud condensation nuclei (CCN). NPF is an objective phenomenon and therefore its contribution to CN and CCN is definitely positive. "The nucleation resulted in decreased CCN" is only based on the simulation analysis. The authors should be cautious about the discussions relevant to "negative contribution"

Thanks for the suggestions. We have rephrased the discussions by specifically clarifying the results are from simulations.

(2) When using empirical formulation to calculate nucleation, the overestimation of CN could not be simply attributed to the high prefactor/coefficient in formulation. The empirical formulation has physical flaws because it could not consider the influence of other factors besides sulfuric acid, e.g., temperature and condensation sink.

We agree that temperature and condensation sink are two key parameters affecting nucleation processes, primarily by influencing the level of gaseous precursors involved in the nucleation, and thus the particle number concentration (Zhu et al., 2017). The empirical nucleation parameterizations used in this study is strongly governed by the nucleation coefficient, which indeed is a combination by taking into account the physical and chemical properties of the nucleation process under different environments, including temperature. Some studies have attempted to lower the

nucleation coefficient by an order of magnitude so as to reduce the overestimation of particle number concentration, but overestimation still exists (Matsui et al., 2013; Arghavani et al., 2022; Dong et al., 2019). In contrast, other studies have pointed out the overestimation of particle number concentration is likely induced by too high nucleation precursors such as sulfuric acid (Cai et al., 2016; Matsui et al., 2011). The numerical experiment supported their hypothesis, indicating that accurate simulation of sulfuric acid plays pivotal roles in improving particle number concentrations.

In terms of the reviewer's concern about temperature, previous studies have tested some nucleation schemes by including temperature, and they found that temperature plays a large role when it is high such as over summer, but plays a marginal role under low temperature conditions. Regarding condensation sink, it is taken into account in the current simulations. Specifically, the Model for Simulating Aerosol Interactions and Chemistry (MOSAIC) used in this study include dynamic gas-particle mass transfer to represent the condensation growth of aerosol, during which the effect of condensation sink of gases precursors on particles ensures a reasonable level of gas precursors involved in nucleation.

(3) The model bias in this study is only based on the comparison of simulation against measurements at one coastal site. Some discussions may be not applicable to other regions. I suggest that the authors can do some work based on more measurements if possible. At least, discussions on existing modeling studies in China or the influence of factors affecting model performance in other regions of China should be added.

Following your comments, we added two more sites for evaluation, including one site over urban Beijing and the other over the rural area of Gucheng. In addition to Qingdao, we further evaluate the performance of WRF-Chem with updated parameterization of the particle formation and growth processes in reproducing the observed particle number concentrations over a few other sites in the North China Plain, including one site over urban Beijing and the other one over the rural area of Gucheng (see methods in the manuscript). In February 2017, there are 10 and 5 NPF events occurred in Beijing and Gucheng, respectively. The model evaluation based on these two sites in general supports the findings over the site of Qingdao. Specifically, the

simulations using activation-type nucleation mechanism with the mass accommodation coefficient of 0.1 (red lines in Fig. 1), the same as Base in section 3.2.1 in the manuscript, substantially overestimates the number concentration of particles in 10–40 nm. The mean fractional bias of $CN_{10-40}$ in Beijing and Gucheng are 81% and 62% respectively, which is strongly reduced to 23% and 11% by increasing the mass accommodation coefficient of sulfuric acid to 0.65 (see Section 3.2.1 of the manuscript for details).

`

Figure 1 The time series of $CN_{10-40}$ on NPF days in (a) Beijing and (b) Gucheng on February 5-24, where red and blue represent Base and MAC simulation results respectively. All times are local times (LT).

For the larger particles (40–100 nm) which are greatly affected by the condensation process, the relevant parameters are also adjusted. For instance, the modified process includes the amount of nitrate condensation in particles below 40 nm and the emission phase of primary organic aerosol (section 3.2.2 of the manuscript), and the yield of SI-SOA in the model (named Low_yield, see section 3.3 of the manuscript for details). For activation-type nucleation mechanism, the mean fractional bias is reduced from 103% to 59% in Beijing, 50% to -5% in Gucheng, with correlation coefficient increasing from 0 to 0.49 and 0.46. (Figure 2).

[Figure]

Figure 2 The time series of $CN_{40-100}$ on NPF days in (a) Beijing and (b) Gucheng on February 5-24, where red and blue represent Base and Low_yield simulation results respectively. All times are local times (LT).

(4) The comparison of the simulated mass concentration of aerosol components and gaseous precursors against observations is encouraged to present before evaluating the model performance in CN.

Following your comments, we have added more model evaluations, including $PM_{2.5}$ compositions derived from a near real-time air pollutant database (Tracking Air Pollution in China, http://tapdata.org.cn) (Geng et al., 2017; Liu et al., 2022) such as $SO_4^{2-}$, $NO_3^-$, $NH_4^+$, OM (Fig. 3) and criteria air pollutants, including particulate matter ($PM_{2.5}$ and $PM_{10}$) and gaseous pollutants ($O_3$, $SO_2$, CO, and $NO_2$) (Fig. 4) in the megacity of Beijing and a coastal city of Qingdao during the simulation period. Overall, relatively low bias is achieved for most of the species.

[Figure]

Figure 3 The comparison between model simulations (red lines) and observations (black lines) from February 5 to February 24, 2017. Shown are results of the average daily concentration of the four main components of $PM_{2.5}$ ($SO_4^{2-}$, $NO_3^-$, $NH_4^+$, OM) in Qingdao (top) and Beijing (bottom). Statistical indicators including mean fractional bias (MFB), mean fractional error (MFE) and correlation coefficient (R) are also displayed in the upper left corner of each panel.

[Figure]

Figure 4 The comparison between model simulations and observations from February 5 to February 24, 2017. Shown are results of concentrations of air pollutants ($O_3$, $SO_2$, CO $NO_2$, $PM_{10}$ and $PM_{2.5}$) in Qingdao (Fig. 5a–f) and Beijing (Fig. 5g–l). Statistical indicators including mean fractional bias (MFB), mean fractional error (MFE) and correlation coefficient (R) are also displayed in the upper left corner of each panel.

(5) The method of calculating CCN is necessary in Sect.2.

According to your comments, we have added the calculation method of CCN in the manuscript. The calculation method of CCN concentration in the WRF-chem model is based on the study of Matsui et al. (2011). According to Köhler theory, CCN concentrations under three given supersaturations of 0.2%, 0.4% and 0.6% were calculated. The critical supersaturation ($S_c$) of each size bin in the WRF-chem model was calculated by the following formula:

$$S_c = \sqrt{\frac{4 \times a^3}{27 \times r^3 \times \kappa}} \tag{1}$$

$$a = \frac{2 \times \sigma}{R_v \times T \times \rho_\omega} \tag{2}$$

Where α (m) is the coefficient of the Kelvin effect, κ is the volume−averaged hygroscopicity, calculated using the values in Table 1, r (m) is the dry diameter, σ is droplet surface tension over water (0.076 N m$^{-1}$), $R_v$ is the gas constant for water vapor (461.6 J K$^{-1}$kg$^{-1}$), T (K) is the air temperature, and $\rho_\omega$ is the density of water (1000 kg m$^{-3}$).

Table 1 Hygroscopicity Parameters (κ) in the WRF-Chem Model

| Species | Hygroscopicity (κ) |
| --- | --- |
| Sulfate | 0.5 |
| Ammonium | 0.5 |
| Nitrate | 0.5 |
| Black carbon | $10^{-6}$ |
| Primary organic aerosol | 0.14 |
| Other inorganics | 0.14 |
| Sodium | 1.16 |
| Chloride | 1.16 |

(6) How is the emission of IVOC considered?

In the 2-species volatility basis set (VBS) treatment of secondary organic aerosol formation used in WRF-chem in this study, SVOC and IVOC emissions corresponding

to both anthropogenic and biomass burning emissions are derived based on a constant emission ratio of S/IVOC to POA (Shrivastava et al., 2011). Specifically, the emission of IVOC is assumed to be 6.5 times that of POA, which may have some uncertainty due to different emission ratios of POA from different source, requiring more work in future for the investigation.

(7) Is Knudsen effect considered when calculating the condensation of gases onto nano-size particles?

Yes, the model used in the study takes into account the Knudsen effect when calculating the condensation of gases on particles. Gas-particle partitioning process for various species are delineated using the Model for Simulating Aerosol Interactions and Chemistry scheme (MOSAIC) in the WRF-Chem model. The following formula is used to calculate first order mass transfer coefficient for gaseous precursors on particles, where Knudsen effect is taken into account (Zaveri et al., 2008):

$$K_{i,m} = 4\pi \bar{R}_{p,m} D_{g,i} N_m \mathrm{f}\left(Kn_{i,m}, \alpha_i\right) \tag{3}$$

$$\mathrm{f}\left(Kn_{i,m}, \alpha_i\right) = \frac{0.75\alpha_i\left(1 + Kn_{i,m}\right)}{Kn_{i,m}\left(1 + Kn_{i,m}\right) + 0.283\alpha_i\, Kn_{i,m} + 0.75\alpha_i} \tag{4}$$

Where $k_{i,m}$ ($s^{-1}$) is the first order mass transfer coefficient for species i and bin m; $\bar{R}_{p,m}$ (cm) is mean wet radius of particles in bin m; $D_{g,i}$ ($cm^2\ s^{-1}$) is gas diffusivity of species i; $N_m$ ($cm^{-3}$) is the number concentration of particles in bin m; $f(Kn_{i,m}, \alpha_i)$ is the transition regime correction factor to the Maxwellian flux as a function of the Knudsen Number (Fuchs and Sutugin, 1971); $\alpha_i$ refers to the mass accommodation coefficient.

(8) The simplified SOA simulation in this study did not consider the multi-stage oxidation of organic species, which can affect the microphysical properties of organic species in particle growth processes. Discussions on this issue is suggested.

Thank you for your valuable comments. The simplified VBS mechanism used in our study does not consider the multi-step oxidation of organic species, and we have added some elaboration to address the uncertainties.

In the 2-species VBS mechanism used in the study, SI-SOA with effective saturation concentrations ($C^*$) of $10^{-2}$ µg $m^{-3}$ is formed by the one-step vapor phase

oxidation of S/IVOC vapors with $C_*$ of $10^5$ μg m$^{-3}$ through a reduction of volatility by 7 orders of magnitude. Instead of representing the real physics, this process aims to parameterize the mean effect of the complex processes of SOA formation (Shrivastava et al., 2011), which may potentially underestimate SOA (Chrit et al., 2018; Zhao et al., 2020) concentrations and the subsequent effect on CCN.

(9) The model resolution is not high enough to well resolve the urban pollution. In discussing the model performance, this factor should be taken into account.

We understand that a higher spatial resolution may be desirable when urban pollution is a major concern. In the revised manuscript, we have added two more sites (one over urban and the other over rural area) based on the reviewer's earlier comments, and the model in general shows reasonable performance. We do believe higher spatial resolution simulations may improve the performance if the spatial resolution of emission inventory improves. We have incorporated the factor of resolution into the Method section in the revised manuscript.

(10) Fig.3 shows a different temporal variation of CN$_{10-40}$ from that of CN$_{40-100}$ which shares a similar pattern with that of CN$_{100-1000}$. Primary emission may be a dominant factor contributing to model bias in CN$_{40-100}$. Is the diurnal variation of emission considered in emission data? Correlation coefficient between the simulation results and the observations is recommended to show.

Diurnal variations in emissions are taken into account when processing emission data in the WRF-chem model. Based on the reviewer's comments, we calculated the correlation coefficient and added into Table S1 and S2 in the supporting information.

**Technical suggestions:**

L191 and L193, organic aerosols -> organic matters

Done

L269, NFP-explicit -> NPF-explicit

Done

In Fig.2 and Fig.3, horizontal label bar is better

Done

Higher and narrower Fig.4 with horizontal label bar is better.

The figure and label have been revised.

In fig.5, a unified color bar for fig.5a, b, and c would be better.

Done

**References:**

Arghavani S, Rose C, Banson S, et al. 2022. The Effect of Using a New Parameterization of Nucleation in the WRF-Chem Model on New Particle Formation in a Passive Volcanic Plume. Atmosphere [J], 13(1): 15.

Cai C, Zhang X, Wang K, et al. 2016. Incorporation of new particle formation and early growth treatments into WRF/Chem: Model improvement, evaluation, and impacts of anthropogenic aerosols over East Asia. Atmospheric Environment [J], 124(262-284.

Chrit M, Sartelet K, Sciare J, et al. 2018. Modeling organic aerosol concentrations and properties during winter 2014 in the northwestern Mediterranean region. Atmos. Chem. Phys. [J], 18(24): 18079-18100.

Dong C, Matsui H, Spak S, et al. 2019. Impacts of New Particle Formation on Short-term Meteorology and Air Quality as Determined by the NPF-explicit WRF-Chem in the Midwestern United States. Aerosol and Air Quality Research [J], 19(2): 204-220.

Fuchs N A, Sutugin A G 1971. HIGH-DISPERSED AEROSOLS [M] //G. M. HIDY, J. R. BROCK, Topics in Current Aerosol Research. Pergamon: 1.

Geng G, Zhang Q, Tong D, et al. 2017. Chemical composition of ambient PM2. 5 over China and relationship to precursor emissions during 2005–2012. Atmos. Chem. Phys. [J], 17(14): 9187-9203.

Liu S, Geng G, Xiao Q, et al. 2022. Tracking Daily PM2.5 Chemical Composition in China Since 2000 Based on Multisource Data Fusion [M]: A45B-04.

Matsui H, Koike M, Kondo Y, et al. 2011. Impact of new particle formation on the concentrations of aerosols and cloud condensation nuclei around Beijing. Journal of Geophysical Research: Atmospheres [J], 116(D19).

Matsui H, Koike M, Takegawa N, et al. 2013. Spatial and temporal variations of new particle formation in East Asia using an NPF-explicit WRF-chem model: North-south contrast in new particle formation frequency. 118(20): 11,647-611,663.

Shrivastava M, Fast J, Easter R, et al. 2011. Modeling organic aerosols in a megacity: comparison of simple and complex representations of the volatility basis set approach. Atmospheric Chemistry and Physics [J], 11(6639-6662.

Zaveri R A, Easter R C, Fast J D, et al. 2008. Model for Simulating Aerosol Interactions and Chemistry (MOSAIC). 113(D13).

Zhao B, Shrivastava M, Donahue N M, et al. 2020. High concentration of ultrafine particles in the Amazon free troposphere produced by organic new particle formation. Proceedings of the National Academy of Sciences [J], 117(41): 25344-25351.

Zhu Y, Yan C, Zhang R, et al. 2017. Simultaneous measurements of new particle formation at 1 s time

resolution at a street site and a rooftop site. Atmos. Chem. Phys. [J], 17(15): 9469-9484.

---

## Author Comment (AC2)

We thank the reviewer for the comprehensive comments to help us improve the manuscript. Please see the detailed responses to your comments below.

The motivation behind this study is to update the nucleation and growth parameterizations in the WRF-Chem model, enabling it to simulate the particle formation and CCN formation processes in a coastal city in China. While some earlier studies have found a negative contribution of NPF to CCN, this study finds a positive contribution of NPF to CCN by adjusting the SI-SOA yield. The major updates include changes to key parameters, such as the $H_2SO_4$ accommodation rate, the $HNO_3$ condensation rate, the direct emission of primary organic aerosol, and, most importantly, the SI-SOA yield. This type of work is encouraged and fits within the scope of ACP.

[disclaimer: I'm not an expert who can judge whether the authors' model and setups represent the most advanced knowledge in their community.]

**Major comments:**

1. The authors should directly change the mass accommodation coefficient of $H_2SO_4$ from 0.1 to 1. There is enough experimental evidence showing this is the case. All other results should be revised with respect to this change. See 10.5194/acp-20-7359-2020

Thanks for the suggestions. In order to incorporate this suggestion and address the comments below, we have added another scheme (kinetics) in which the mass accommodation coefficient of $H_2SO_4$ is set to 1.0. While completely repeating all experiments would take tremendous of time and computational resources, we have added a number of simulations for the model evaluation and comparison. Therefore, the revised layout of the manuscript is to keep the original structure, but add the discussions based on kinetics scheme at the end of the manuscript.

2. L263: I believe it's time for everyone to stop using the activation scheme, given that the studies supporting it are from 2006-2007 and even the authors themselves may have moved on. Moreover, the dependence of J on $H_2SO_4$ is evidently non-linear. Several

studies in Chinese megacities have demonstrated the significance of $H_2SO_4$-DMA nucleation. While the situation might be different in a coastal city, it's unlikely that there is no $NH_3$ present. Incorporating DMA and $NH_3$ into WRF-chem may be challenging, as their sources may not be explicitly described. However, I encourage the authors to employ the $H_2SO_4$- $NH_3$ nucleation mechanism and rates in their study. They could use an estimated $NH_3$ concentration, as the nucleation rate from H2SO4-NH3 is less dependent on $NH_3$ than on H2SO4. The authors should compare the results obtained using the activation scheme with those obtained using the H2SO4-NH3 mechanism. If the latter yields superior results, it should be used as the default for other sensitivity tests. Conversely, if the $H_2SO_4$-$NH_3$ mechanism does not improve the results, this issue should be discussed. The problem may lie in other less certain modules instead of this experimentally confirmed mechanism.

Thanks for the suggestions and comments. As was described in the response to the first comment, we have added a number of new simulations. For instance, considering the reviewer's concern about the nonlinearity of the dependence of J on $H_2SO_4$, we added another nucleation scheme of kinetics, which assumes that the nucleation rate is proportional to the square of the concentration of sulfuric acid ($J = K[H_2SO_4]^2$). In this scheme, the mass accommodation coefficient of $H_2SO_4$ is set to one, and all the adjustment discussed in the linear-$H_2SO_4$ has been added in this scheme. The resulting simulated results are comparable to those obtained by the linear-$H_2SO_4$ nucleation mechanism. In addition, we conducted another set of simulations with $H_2SO_4$-$NH_3$ nucleation scheme, and the comparison indicates that the simulations under this scheme substantially overestimate the particle number concentrations. The reason likely lies in, that the reviewer has mentioned, the $H_2SO_4$-DMA was recently proposed to be the major nucleation scheme in megacities of China. To this end, we have added the relevant discussions in supplementary section S2. The section S2 is shown below as well.

To further verify the robustness of the model improvement in reproducing the observations, we select another empirical scheme, e.g., kinetics, nucleation for

evaluation. The repeated analysis for the smaller particle number concentrations ($CN_{10-40}$) indicates comparable performance between kinetics and activation schemes (Fig. S9), both showing improvement when mass accommodation coefficient is increased from 0.1 to 0.65. Considering that the mass accommodation coefficient is suggested to reach one in some studies (Stolzenburg et al., 2020), we therefore conduct another simulation under the kinetics nucleation scheme by increasing the mass accommodation coefficient to 1.0 (purple lines in Fig. 1), yielding comparable performance but with negative mean fractional bias contrasting to the positive one based on mass accommodation coefficient of 0.65 (green lines in Fig. 1; Table S1). For the large particle number concentrations ($CN_{40-100}$), the adjusted mass accommodation coefficient (1.0) together with low yield of SI-SOA at kinetics scheme shows similar improvements as activation (Fig. 2 and Table S2).

[Figure]

Fig. 1. The time series of $CN_{10-40}$ on NPF days in (a) Qingdao, (b) Beijing and (c) Gucheng on February 5-24 simulated by Base (marked in orange) and MAC (green and purple lines corresponding to sulfuric acid mass coefficient of 0.65 and 1, respectively) using kinetics nucleation scheme (KIN) as well as from observations (OBS) (marked in black). All times are local times (LT).

Table 1 The statistics of model simulation and observation data for $CN_{10-40}$ in Qingdao, Beijing and Gucheng

| Observational sites / Simulation | Qingdao | | | Beijing | | | Gucheng | | |
|---|---|---|---|---|---|---|---|---|---|
| | MFB (%) | MFE (%) | R | MFB (%) | MFE (%) | R | MFB (%) | MFE (%) | R |
| ACT_Base | 48% | 66% | 0.69 | 81 | 90 | 0.35 | 62 | 82 | 0.21 |
| ACT_ MAC(0.65) | 1% | 49% | 0.70 | 23 | 65 | 0.39 | 11 | 67 | 0.13 |
| KIN_Base | 58% | 83% | 0.60 | 86 | 91 | 0.41 | 76 | 93 | 0.13 |
| KIN_ MAC(0.65) | 40% | 71% | 0.60 | 41 | 78 | 0.34 | 37 | 81 | 0.18 |
| KIN_MAC(1.0) | -30% | 57% | 0.69 | -40 | 61 | 0.41 | -34 | 81 | 0.23 |

[Figure]

Fig. 2. The time series of $CN_{40-100}$ on NPF days in (a) Qingdao, (b) Beijing and (c) Gucheng on February 5-24 simulated by Base (marked in orange) and Low_yield (marked in dark green) using kinetics nucleation scheme (KIN) as well as from observations (OBS) (marked in black). All times are local times (LT).

Table 2 The statistics of model simulation and observation data for $CN_{40-100}$ in Qingdao, Beijing and Gucheng.

| Observational sites / Simulation | Qingdao | | | Beijing | | | Gucheng | | |
|---|---|---|---|---|---|---|---|---|---|
| | MFB (%) | MFE (%) | R | MFB (%) | MFE (%) | R | MFB (%) | MFE (%) | R |
| ACT_Base | 98 | 102 | 0 | 103 | 106 | 0 | 50 | 72 | 0 |
| ACT_Lowyield | 32 | 53 | 0.42 | 59 | 65 | 0.47 | -5 | 47 | 0.46 |
| KIN_Base | 88 | 94 | 0 | 97 | 100 | 0 | 50 | 74 | 0 |
| KIN_Lowyield | 36 | 52 | 0.39 | 53 | 60 | 0.49 | -7 | 48 | 0.46 |

Following the empirical nucleation scheme, we then conduct a classical nucleation mechanism to take both chemical species and meteorological conditions directly into account (Sihto et al., 2006). For instance, we select a commonly used $H_2SO_4$-$H_2O$-$NH_3$ ternary homogeneous nucleation which is highly dependent on temperature and relative humidity (Napari et al., 2002). The number concentrations at 10–40 nm are much higher (Fig. 3), at either low or high mass accommodation coefficient, compared to observations and the empirical schemes abovementioned, and the diminished effect during the adjustment of mass accommodation coefficient is likely a result of $NH_3$.

[Figure]

Fig. 3. The time series of $CN_{10-40}$ on NPF days in (a) Qingdao, (b) Beijing and (c) Gucheng on February 5-24 simulated by Base (marked in purple) and MAC (marked in yellow) using $H_2SO_4$-$H_2O$-$NH_3$ ternary homogeneous nucleation (THN) as well as from observations (OBS) (marked in black). All times are local times (LT).

Contrasting to the scheme of $H_2SO_4$-$H_2O$-$NH_3$, the formation of sulfuric acid (SA)-dimethylamine (DMA)-water clusters has been found to be important sources of new particle formation in megacities over China (Yao et al., 2018). Bergman et al. (2015) applied amine-enhanced nucleation parameterization to an aerosol climate model to estimate the effect of amine on new particle formation on a global scale, indicating that high nucleation rates are confined to regions close to the amine source due to the short lifetime of amines. Because of the short life of amines, the emission of amines remains to be highly uncertain and deserves further investigation (Chang et al., 2021). By comparing this classical nucleation scheme with the empirical one (e.g., kinetics), the spatial distibutions of particle formation rate between these two types of nucleation schemes are largely consistent.

3. I find Session 3.4 to be particularly fascinating, but it's currently buried amidst a lot of less significant information. This session should be considered one of the key findings of this study and given prominence in both the abstract and conclusion. The

yield of SI-SOA remains highly uncertain, and I'm surprised to learn that such a small change in the reaction rate coefficient can have such a significant impact on the contribution of NPF to CCN. I hadn't expected this result at all. This finding underscores the need for further research into SI-SOA yield in polluted environments, particularly since urban environments are highly complex and model treatments are often oversimplified. Clearly, a better understanding of NPF's contribution to CCN hinges on a better grasp of this prerequisite knowledge.

We thank the reviewer for the positive comment on the discussion of session 3.4. We have revised the discussion based on the reviewer's suggestion. As the reviewer pointed out the uncertainty, actually when we apply the quadratic-$H_2SO_4$ nucleation scheme, the sensitivity numerical simulations with high and low yield of SI-SOA result in comparable contributions to CCN, which differs from the results using the linear-$H_2SO_4$ nucleation scheme. It may be related to particulate hygroscopicity dependence on the nucleation scheme. More future studies are necessary to investigate this issue. As was suggested by the reviewer, more work is needed to improve understanding of NPF's contribution to CCN.

**Minor comments:**

L39-41: a number of observations may be misleading. There are more than enough observations showing the positive correlation of NPF and CCN. Additionally, while some simulations do not show positive correlation of NPF and CCN in a global scale, many of the models do. If the authors' statement is about e.g., polluted environments or more specific the Chinese city, the authors should clearly be stating so. Otherwise they should modify this sentence properly.

Thanks for the suggestion. We have rephrased the descriptions. The words of "polluted environment" has been added to constrain the descriptions of negative contribution of NPF to CCN.

L74: high-efficiency nucleation efficient nucleation

Done.

L110-121: it appears the authors are only talking about WRF-chem. They should carefully mention this clearly in the manuscript that it is the WRF-chem model, not a general "model" that is observing negative correlation between NPF and CCN.

Done. We have refined the description of the model in the manuscript.

L190: change C* to $c*$ (italic, lower case) throughout the manuscript.

Done.

L199: Please write explicitly the used equation and all the parameters.

Done.

L333: try to reduce using MFB etc. Use the abbreviations only for models runs otherwise readers easily get confused.

We have reduced the abbreviations in the revised manuscript.

L347: space between $\mu g\ m^{-3}$play

Revised.

L349: Does this mean that all the vapours only have two different volatilities?

The volatilities mentioned in this sentence mainly indicate that for primary organic aerosol, with effective saturation concentration ($c*$) of POA is $10^{-2}$ and $10^{5}$ $\mu g\ m^{-3}$, respectively. For gas phase SOA oxidized by volatile organic compounds, the effective saturation concentration is set to be one, which is not the focus of the study, therefore, we did not mention the volatility of this part SOA. To make it clear, we have deleted the descriptions of two sets of volatilities, and directly discuss the source of primary organic aerosol with effective saturation concentration at $10^{-2}$ $\mu g\ m^{-3}$.

L354: Gas phase POA forms close to the emission source. However, with…

Done.

L356: Therefore, POA may not contribute to particle growth away from the emission sources. Or something similar.

The sentence has been elaborated.

L364: Please label the Figure S1 panels. There are also clear signs of grey bars in the figure. Please remove those when putting the figures together.

Done.

L412: Avoid using too many abbreviations (PNC).

The abbreviations have been removed.

L416: Which nucleation mechanism is used in this study?

Organics nucleation mechanism is used in this study, which has been added in the revised manuscript.

L506: Are there measurement data for figures d,e,f?

We have observations for particle number concentrations over these three bins. Since we have done particle number concentration evaluations in Fig. 3, we try not to repeat the information in this figure.

L538: This session is very interesting. I think this is worthy to be emphasised.

Thanks for your suggestion. We have elaborated the writing, and added another scheme (kinetics) to further support the finding.

**References**

Bergman T, Laaksonen A, Korhonen H, et al. 2015. Geographical and diurnal features of amine-enhanced boundary layer nucleation. Journal of Geophysical Research: Atmospheres [J], 120(18): 9606-9624.

Chang Y, Gao Y, Lu Y, et al. 2021. Discovery of a Potent Source of Gaseous Amines in Urban China. Environmental Science & Technology Letters [J], 8(9): 725-731.

Napari I, Noppel M, Vehkamäki H, et al. 2002. Parametrization of ternary nucleation rates for H2SO4-NH3-H2O vapors. Journal of Geophysical Research: Atmospheres [J], 107(D19): AAC 6-1-AAC 6-6.

Sihto S L, Kulmala M, Kerminen V M, et al. 2006. Atmospheric sulphuric acid and aerosol formation: implications from atmospheric measurements for nucleation and early growth mechanisms. Atmos. Chem. Phys. [J], 6(12): 4079-4091.

Stolzenburg D, Simon M, Ranjithkumar A, et al. 2020. Enhanced growth rate of atmospheric particles from sulfuric acid. Atmos. Chem. Phys. [J], 20(12): 7359-7372.

Yao L, Garmash O, Bianchi F, et al. 2018. Atmospheric new particle formation from sulfuric acid and amines in a Chinese megacity. Science [J], 361(278-281.

---

## Author Comment (AC3)

**Response to Reviewer #3** (our response in **colour**)

We thank the reviewer for the constructive comments to help us further improve the manuscript. Please see the detailed responses to your comments below.

**Reviewer #3:**

This study improves the simulation accuracy of particle and CCN number concentration through adjusting serval parameters such as the mass accommodation coefficient. The results are interesting and has a good potential to improve the model of NPF and its impact on CCN number concentration. This study is within the scope of the journal ACP. I recommend this paper for publication after the following issues are resolved.

**Major comments:**

1. Line 147. Li et al (2015) shows the measurement of bulk CCN, not the measurement of size-resolved CCN. The authors should give more information about the size-resolved CCN measurement, including the flow set, multi-charge calibration method, data quality control and so on.

Thank you for your comments. We have revised the observation method of CCN in section 2.1 of the manuscript. The observations of bulk CCN refer to the study of Li et al. (2015).

2. Line 262: The extremely low volatility volatile organic compounds (ELVOCs) can also have a contribution for the nucleation events.

Right, extremely low volatile organic vapors can be involved in particle nucleation. In this study, the empirical nucleation scheme is strongly governed by the nucleation coefficient, which indeed is a combination of multiple chemical species with different volatilities, which should include the extremely low volatility volatile organic compounds.

2. Line 458-460: The activation critical particle size at SS=0.2% is about 120~130 nm. It is interesting that the $CCN_{0.2\%}$ has an obvious underestimation when $CN_{10-1000nm}$ has an overestimation (Figure 3c) for the RACD scheme. The change of activation ratio ($N_{CCN}/N_{CN}$) can be further analyzed.

Thanks for your comments and we add the analysis of the activation ratio. The results are consistent with the trend of CCN, in general showing similar features as CCN.

[Figure]

Fig. 1. Average diurnal variation of (a) $CCN_{0.2\%}$, (b) $CCN_{0.4\%}$ and (c) $CCN_{0.6\%}$ on NPF days in Qingdao on February 5-24, 2017, in Low-yield and High-yield simulations, shown as blue and brown lines. The solid line representing CCN concentration and the dashed line corresponding activation ratio (AR).

The underestimation of CCN at low supersaturation (0.2%) concomitant with the overestimation of particles at 100–1000 nm under high yield SI-SOA, as mentioned by the reviewer, is believed to be influenced by the hygroscopicity of the particles in this study. Compared to the high yield of SI-SOA, the reduced yield of SI-SOA tends to decrease organic matter which enhances the hygroscopicity of the particles to be easier to activate to CCN, therefore, the bias of CCN simulation is substantially reduced. To verify this hypothesis, we calculated the hygroscopicity parameter (kappa) for 100–1000 nm particles, and the values increase from 0.26 at high SI-SOA yield to 0.30 at low SI-SOA yield, indicating the enhanced hygroscopicity at low SI-SOA yield.

1. Line471-473: Why secondary inorganic aerosols (SIA) is not the major contributor of condensational growth?

Different chemical species are involved in the growth of particles. Secondary inorganic aerosols (ammonium, nitrate, sulfate) and secondary organic aerosols all make important contributions to the growth of newly formed particles in the atmosphere (Xiao et al., 2015; Ehn et al., 2014; Zhu et al., 2019). However, their importance differs

dependent on the particle sizes. For particles less than 40 nm, the condensation of semi-volatile substances such as nitric acid on particles tends to be inhibited by the high acidity of small particles (Roldin et al., 2011; Deming and Ziemann, 2021). For instance, based on observed evidence, Li et al. (2022) found in Beijing in 2018 showed that organic matter is the main component of 8−40 nm particles, with mass fraction of 80±8%, followed by sulfates accounting for 13±7%, and inorganic nitrates accounting for less than 3%. Therefore, we emphasize the secondary inorganics may not play a major role for smaller particles.

2. Figure 5. The authors emphasize that the model modification has a largest impact on the simulation of CCN over the North China Plain. It is necessary to verify the result based on the measurement in this region.

Thanks for the suggestions. We have tried to achieve more data, and it turned out we got more data on particle number concentrations, but not CCN. Therefore, we added more evaluations of particle number concentrations over two more sites at North China Plain. The model evaluations (Section S2 in the supporting information) are in general consistent with that in Qingdao, warranting the confidence of the model used in the study.

**Technical suggestions:**

Line 249: "particles growth" should be "particle growth"

Done.

Line 269: "NFP" should be "NPF"

Revised.

Line 316-328: This paragraph use "α" and "MAC" as the abbreviation of "mass accommodation coefficient". I suggest using a unified symbol in text

Done. We have reduced the use of abbreviations in the manuscript.

**References:**

Deming B, Ziemann P 2021. Measurements of the partitioning of nitric acid and sulfuric acid in aqueous/organic phase-separated systems. Environmental Science: Atmospheres [J], 1(

Ehn M, Thornton J, Kleist E, et al. 2014. A large source of low-volatility secondary organic aerosol. Nature [J], 506(476-479.

Li K, Zhu Y, Gao H, et al. 2015. A comparative study of cloud condensation nuclei measured between non-heating and heating periods at a suburb site of Qingdao in the North China. Atmospheric Environment [J], 112(40-53.

Li X, Li Y, Cai R, et al. 2022. Insufficient Condensable Organic Vapors Lead to Slow Growth of New Particles in an Urban Environment. Environmental Science & Technology [J], 56(14): 9936-9946.

Roldin P, Swietlicki E, Schurgers G, et al. 2011. Development and evaluation of the aerosol dynamics and gas phase chemistry model ADCHEM. Atmos. Chem. Phys. [J], 11(12): 5867-5896.

Xiao S, Wang M Y, Yao L, et al. 2015. Strong atmospheric new particle formation in winter in urban Shanghai, China. Atmos. Chem. Phys. [J], 15(4): 1769-1781.

Zhu Y, Li K, Shen Y, et al. 2019. New particle formation in the marine atmosphere during seven cruise campaigns. Atmos. Chem. Phys. [J], 19(1): 89-113.

---

## Referee Report (RR1)

The authors answered the main questions raised by reviewers and the revised manuscript is well improved. To make the readers get more understanding on this study, I recommend the authors can added some quantitative discussions based on the observations in China and simulation studies using other models and model configurations where necessary before the paper is accepted for publication.

---

## Author Response (AR2)

**Response to Reviewer #1** (our response in **colour**)

We thank the reviewer for the constructive comments to help us further improve the manuscript. Please see the detailed responses to your comments below.

The authors answered the main questions raised by reviewers and the revised manuscript is well improved. To make the readers get more understanding on this study, I recommend the authors can added some quantitative discussions based on the observations in China and simulation studies using other models and model configurations where necessary before the paper is accepted for publication.

Thanks for the suggestions. We have added several recent observational studies on the chemical composition of ultrafine particles and CCN in China, as well as related simulation studies in the manuscript. Please see the detailed information below, as well as in the revised manuscript.

Lines 419-422:

"*Recent observations conducted in Beijing also indicated that particles at 8–40 nm are mainly composed of organic matter (with mass fraction of ~80%) and sulfate (with mass fraction of ~13%), while nitrate content is very low (with mass fraction of ~3%) (Li et al., 2022)*"

Lines 566-570:

"*Similarly, based on observational data in northern China in summer, Wang et al. (2023) found that CN in 2020 is lower than that in 2014 due to particulate pollution control, however, the particles become more easily activated, attributable to the larger extent of decrease in organic matters compared to inorganics, leading to enhanced particle hygroscopicity and more conducive to activation.*".

Lines 605-608:

"*For instance, by adding the formation chemistry associated with multi-generational oxidation, Zhao et al. (2020) found improved simulations of vertical aerosol profile in the Amazon free troposphere compared to the simplified VBS mechanism.*".

**Response to Reviewer #2** (our response in **colour**)

We thank the reviewer for the constructive comments to help us further improve the manuscript. Please see the detailed responses to your comments below.

1. It's not clear what the last sentence "The substantial contribution of new particle formation to CCN under low SS and SI-SOA is applicable to other mechanisms such as kinetics." is talking about. Please rewrite it to clarify the message.

Thanks for the suggestions, and the discussion of contribution of new particle formation to CCN under other mechanisms has been added in the third paragraph of the conclusions.

2. The author's new simulations by adopting the kinetic nucleation rate suggest that both high and low SI-SOA schemes yield a positive contribution of NPF to CCN. This is distinct from using the out-of-date activation scheme which suggested the high SI-SOA yield was the problem causing the under-estimation of NPF to CCN.

Therefore, the question becomes: is the SI-SOA the problem or the nucleation mechanism the problem causing the estimation of NPF to CCN in models? I think the authors may have realised this potential (major) problem but so far I see insufficient discussions about SI-SOA yield vs. nucleation mechanisms. The paper has not changed its root which states SI-SOA yield is the core of the problem (which I'm not so sure about anymore).

However, I agree with the authors that this subject is still very open and more work is needed to tackle the uncertainties. So I would accept the manuscript if the authors properly formulate that the NPF to CCN contribution is affected both by SI-SOA yield and the adopted nucleation mechanisms in their manuscript. They should highlight this key information both in their abstract and conclusions instead of just adding one sentence which does not reflect their new findings during the revisions.

Thanks for the suggestions, and we have added the relevant discussions in the revised

abstract and conclusions (third paragraph). Please see the detailed information below, as well as in the revised manuscript.

Lines 57-60

*"The bias-corrected model is robustly applicable to other schemes, such as quadratic-$H_2SO_4$ nucleation scheme, in terms of CN and CCN, though the dependence of CCN on SI-SOA yield is diminished likely due to changes in particle composition."*

Lines 687-696

*"In addition to activation nucleation scheme, we have also tested a few other schemes such as the quadratic-$H_2SO_4$ nucleation scheme (e.g., kinetics nucleation scheme). Under this scheme, the bias-corrected method abovementioned is applicable to improving the simulations of concentrations of CN and CCN. It is noteworthy that the dependence of CCN on the SI-SOA yield is diminished, showing that under both high and low yields of SI-SOA, there are positive contributions of NPF to CCN. This is likely due to the increase in the amount of sulfuric acid involved in nucleation, making it more hygroscopic and easier to activate to CCN, and the high content of inorganic species makes them less sensitive to changes in SI-SOA yield, which deserves further investigation."*